# TEST-TIME ALIGNMENT FOR LARGE LANGUAGE MODELS VIA TEXTUAL MODEL PREDICTIVE CONTROL

**Kuang-Da Wang**[1,*]   **Teng-Ruei Chen**[1,*]   **Yu-Heng Hung**[1]   **Guo-Xun Ko**[1]
**Shuoyang Ding**[2]   **Yueh-Hua Wu**[2]   **Yu-Chiang Frank Wang**[2,3]   **Chao-Han Huck Yang**[2]
**Wen-Chih Peng**[1]   **Ping-Chun Hsieh**[1]
[1]National Yang Ming Chiao Tung University, Hsinchu, Taiwan   [2]NVIDIA
[3]National Taiwan University, Taipei, Taiwan
{gdwang.cs10,pinghsieh}@nycu.edu.tw, hucky@nvidia.com

## ABSTRACT

Aligning Large Language Models (LLMs) with human preferences through fine-tuning is resource-intensive, motivating lightweight alternatives at test time. We address test-time alignment through the lens of sequential decision making, a perspective that reveals two fundamental challenges. When actions are defined at the token level, as in guided decoding, alignment suffers from the *curse of horizon*. Conversely, when actions are at the response level, as in traditional iterative refinement, the *curse of dimensionality* emerges. To resolve this trade-off, we draw inspiration from Model Predictive Control (MPC) in control theory to propose **Textual Model Predictive Control (TMPC)**, a novel predictive planning framework adapted for aligning LLMs at inference time. A key limitation of standard MPC is its reliance on predefined, hard segment boundaries, which are often absent in text generation. TMPC overcomes this by introducing two principles inspired by hierarchical reinforcement learning: (1) *Hindsight Subgoal Identification*, where TMPC analyzes generation subgoals to retrospectively identify high-reward intermediate outputs as subgoals. This allows the framework to discover meaningful, task-specific planning steps (*e.g.,* a sentence in machine translation or a bug fix in code generation.). (2) *Subgoal-Conditioned Re-Generation*, where these identified subgoals are used to guide subsequent planning iterations. By conditioning on these proven, high-quality subgoals, TMPC ensures stable improvement by building upon previously validated successes. TMPC is evaluated on three tasks with distinct segmentation properties: discourse-level translation, long-form response generation, and program synthesis. The results demonstrate that TMPC consistently improves performance, highlighting the generality. Project page: https://rl-bandits-lab.github.io/TMPC/.

## 1 INTRODUCTION

The emergence of Large Language Models (LLMs), such as the GPT series (Achiam et al., 2023; Brown et al., 2020), LLaMAs (Touvron et al., 2023a;b), and Gemma (Team et al., 2024), has demonstrated remarkable efficacy in a wide range of NLP tasks (Hendrycks et al., 2021; Srivastava et al., 2023; Stiennon et al., 2020; Yu et al., 2024; Zhong et al., 2024). While these models exhibit strong performance out of the box, aligning their outputs to human preferences remains critical, especially for smaller-scale LLMs. For instance, in machine translation (Alves et al., 2024), smaller LLMs (*e.g.,* under 10B parameters) frequently suffer from omissions and semantic drift (Wu et al., 2024). Thus, aligning LLM outputs to preferences remains an essential yet challenging problem.

Training-time approaches such as Reinforcement Learning with Human Feedback (RLHF) (Ouyang et al., 2022) and Direct Preference Optimization (DPO) (Rafailov et al., 2023) have achieved strong results in aligning preferences. However, these methods are resource-intensive and require costly retraining whenever preferences or tasks change. This has spurred interest in *test-time alignment*, where outputs are adapted without updating model parameters, using test-time strategies such as

---

*Equal contribution.

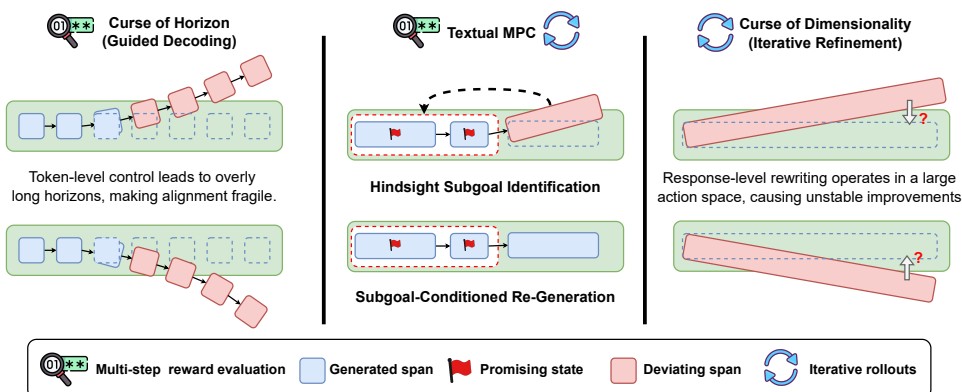

Figure 1: Textual Model Predictive Control (TMPC) balances the curse of horizon in guided decoding against the curse of dimensionality in naive iterative refinement. It employs Hindsight Subgoal Identification to dynamically discover promising states from rollouts and Subgoal-Conditioned Re-Generation to guide the search from these discovered subgoals, ensuring a stable alignment.

prompting (Lin et al., 2024), guided decoding (Khanov et al., 2024; Kong et al., 2024; Li et al., 2024; Wang et al., 2024b; Xu et al., 2025), or iterative refinement (Li et al., 2025).

We address test-time alignment through the lens of sequential decision making, where the generation process is framed as a sequence of actions. This perspective reveals two fundamental challenges, illustrated in Figure 1 . When actions are defined at the token level (*e.g.,* guided decoding), methods suffer from the *curse of horizon* (Park et al., 2025); credit assignment becomes unreliable over long trajectories, making alignment brittle. In contrast, when actions are at the response level (*e.g.,* iterative refinement), they face the *curse of dimensionality*; each step involves rewriting an entire sequence, making the search for improvements in a vast action space intractable and unstable.

To address these challenges, we propose Textual Model Predictive Control (TMPC), a novel test-time alignment framework inspired by Model Predictive Control (MPC) (Camacho & Bordons, 2007; Kouvaritakis & Cannon, 2015). While powerful, standard MPC assumes the problem can be decomposed into predefined, hard segment boundaries, a condition that rarely holds for complex text generation. TMPC is uniquely adapted to overcome this limitation through two principles:

- **Hindsight Subgoal Identification:** This principle allows TMPC to discover meaningful planning steps. After generating candidate responses, TMPC retrospectively analyzes them to identify high-quality intermediate points as *subgoals*. A subgoal can be a concrete unit, such as a sentence in translation, or an abstract one, such as resolving a single failed test case in program synthesis, successfully addressing the problem of lacking natural boundaries. This hindsight-driven discovery effectively shortens the planning horizon for diverse tasks.

- **Subgoal-Conditioned Re-Generation:** This principle ensures stable, cumulative progress. The subgoals identified via hindsight are stored in a buffer and used to guide subsequent planning iterations. By conditioning the next generation on these subgoals, TMPC ensures that subsequent generation builds upon these validated waypoints.

We evaluate TMPC on three challenging tasks with different boundary characteristics: WMT'24 discourse-level machine translation, the HH-RLHF long responses subset, and MBPP program synthesis. Experiments with LLaMA-3.1-8B-Instruct show that TMPC consistently improves alignment, highlighting the generality of our approach.

Our contributions are summarized as follows:

- We formulate test-time alignment under a sequential decision-making lens. This perspective reveals that guided decoding faces a *curse of horizon*, while iterative refinement faces a *curse of dimensionality*. Building on this formulation, we introduce an novel trajectory-

optimization framework for test-time alignment, which, to our knowledge, has not been explored in prior work.

- We introduce Textual Model Predictive Control (TMPC), a framework that adapts concepts from control theory to language generation. TMPC is operationalized through two principles: *Hindsight Subgoal Identification* to discover subgoals from rollouts, and *Subgoal-Conditioned Re-Generation* to iteratively improve generation by building on subgoals.
- We empirically demonstrate the effectiveness of TMPC. TMPC achieves substantial improvements across three distinct domains including long-form response generation, discourse-level machine translation, and program synthesis, validating its ability to discover and leverage task-specific subgoals.

## 2 RELATED WORK

### 2.1 PREFERENCE ALIGNMENT THROUGH FINE-TUNING

Aligning large language models (LLMs) with human preferences has traditionally relied on post-training strategies. Supervised fine-tuning (SFT) (Ziegler et al., 2019) and reinforcement learning from human feedback (RLHF) (Ouyang et al., 2022) (*e.g.,* through Proximal Policy Optimization (Schulman et al., 2017; Huang et al., 2024)) are widely used but computationally expensive. Direct Preference Optimization (DPO) (Rafailov et al., 2023) simplifies RLHF by converting it into a supervised learning objective, though it requires managing dual policies. More recent approaches like SimPO (Meng et al., 2024) and Contrastive Preference Optimization (CPO) (Xu et al., 2024) reduce memory and resource demands using reference models and contrastive signals. Despite these improvements, fine-tuning methods remain rigid and slow to adapt to changing data or objectives, posing challenges in dynamic environments.

### 2.2 TEST-TIME PREFERENCE ALIGNMENT

Test-time preference alignment offers an efficient way to align frozen language models by guiding generation at inference, without requiring any parameter updates. Beyond simple prompting or in-context learning, guided decoding methods harness external signals to control the generation itself. ARGS (Khanov et al., 2024) is a representative example that incorporates reward model guidance at the token level, and InferAligner (Wang et al., 2024b) adopts a similar strategy. Among guided decoding methods, there are also approaches that directly modify internal representations. For instance, RE-Control (Kong et al., 2024) trains a value function on hidden states using the Bellman equation, and applies gradient-based optimization to align with preferences. TreeBoN (Qiu et al., 2025) and RAIN (Li et al., 2024) leverage tree-based structures: TreeBoN combines tree search with Best-of-N sampling, while RAIN performs self-evaluation without relying on a reward model to align preferences. GenARM (Xu et al., 2025) enhances test-time alignment by introducing an autoregressive reward model that predicts next-token reward signals conditioned on prior context, enabling efficient, token-wise guidance that is theoretically expressive under a KL-regularized RL framework. Test-Time Preference Optimization (TPO) (Li et al., 2025) takes a distinct approach, translating reward feedback into textual critiques that serve as language-based rewards. The model uses these to iteratively refine its output—effectively learning alignment on the fly.

### 2.3 SUBGOAL-BASED PLANNING

A long line of work has demonstrated that decomposing a complex task into intermediate *subgoals* can substantially improve long-horizon reasoning, exploration efficiency, and credit assignment (Ren et al., 2019). Language-model-based planners (Logeswaran et al., 2022) generate candidate subgoal sequences to reduce search complexity, while hierarchical RL frameworks for vision-and-language navigation (Wang et al., 2025a) discover intrinsic subgoals that guide agents through sparse-reward environments. Across these settings, subgoals act as local waypoints that stabilize planning and make long-range objectives more tractable. TMPC leverages the same intuition. We leverages intermediate waypoints to simplify long-form generation by uniquely identifying subgoals purely in hindsight from model rollouts at test time with a frozen LLM, offering a single, task-agnostic mechanism across translation, response generation, and program synthesis, unlike prior work relying on explicit environments or learned policies.

## 3 BACKGROUND

In the general setup of RL for LLMs, text generation can be formally modeled as a finite-horizon Markov Decision Process (MDP). We adopt a general notion of a *step* as the basic unit of temporal progression, which can represent a token, a segment at various granularities (*e.g.,* phrase, sentence, or paragraph), or other linguistically or structurally meaningful units. Then, an MDP can be defined as $\mathcal{M} = (\mathcal{S}, \mathcal{A}, \mathcal{P}, R, \mu, T)$, where (i) the state space $\mathcal{S}$ consists of all possible text prefixes, (ii) the action space $\mathcal{A}$ corresponds to the set of all possible generation units, (iii) $\mathcal{P}$ denotes the transition function, (iv) $R : \mathcal{S} \times \mathcal{A} \to \mathbb{R}$ is the reward function that assigns scalar feedback to step-level or trajectory-level outcomes (*e.g.,* measuring fluency, factuality, or alignment with user preferences), (v) $\mu$ denotes the initial state distribution, and (vi) $T \in \mathbb{N}$ is the episode length.

We define the initial state $s_0$ as the initial prompt and let $a_t$ denote the partial response generated at step $t$. At each step $t$, the current state is the set of tokens from the initial prompt and the partial responses generated up to step $t$, *i.e.,* $s_t = (s_0, a_1, \cdots, a_{t-1})$. Based on this construction, we know that the transition function is deterministic with $\mathcal{P}(s_{t+1}|s_t, a_t) = 1$. A policy $\pi_\theta(a|s)$, parameterized by the language model, defines a probability distribution over actions given the prefix $s \in \mathcal{S}$. The generation of a full text sequence of length $T$ can therefore be viewed as a trajectory $\tau = (s_0, a_0, \cdots, s_{T-1}, a_{T-1}, s_T)$ with the cumulative reward given by $\mathcal{J}(\tau) := \sum_{t=0}^{T-1} R(s_t, a_t)$.

This perspective enables the application of RL methods to text generation in LLMs. Rather than relying solely on maximum likelihood estimation, which optimizes local token-level likelihoods, the MDP formulation allows optimization with respect to long-horizon objectives such as coherence and alignment with human preferences. This provides the foundation for recent advances in preference-based fine-tuning and test-time alignment.

## 4 METHODOLOGY

### 4.1 TEST-TIME ALIGNMENT VIA TRAJECTORY OPTIMIZATION

Our key idea is to take a model-based RL viewpoint to achieve test-time alignment for LLMs. Specifically, we propose to recast preference alignment as *trajectory optimization* and thereby employ receding-horizon control for iterative text generation.

**Text Generation Optimization as Trajectory Optimization.** Usually adopted by the model-based RL literature (Chua et al., 2018; Lowrey et al., 2019), the goal of trajectory optimization is to find an optimal sequence of actions $\boldsymbol{a}^* = (a_0^*, \cdots, a_{T-1}^*)$ such that the total trajectory-wise reward is maximized. This matches the objective of LLM text generation in that the output response is generated to best align with the underlying preference. Recall from Section 3 that we adopt a general notion of a *step* as the basic unit of temporal progression, which can be a segment at various granularities or other linguistically meaningful units. Again, we let $s_0$ denote the initial prompt and let $\tau = (s_0, a_0, \cdots, s_{T-1}, a_{T-1}, s_T)$ denote a trajectory generated under an action sequence $\boldsymbol{a}_{0:T-1} := (a_0, a_1, \ldots, a_{T-1})$. Given an initial prompt $s_0$, the search for an optimal sequence $\boldsymbol{a}^*(s_0)$ can be formulated by the following optimization problem

$$\boldsymbol{a}^*(s_0) := \arg \max_{\boldsymbol{a}_{0:T-1}} \sum_{t=0}^{T-1} R(s_t, a_t). \tag{1}$$

Note that there is no need to take expectation in (1) as the state transitions are deterministic given $\boldsymbol{a}_{0:T-1}$ in MDPs for text generation, as described in Section 3.

**Textual Model Predictive Control for Text Generation.** In general, direct optimization of (1) requires searching over all possible action sequences of length $T$ and is computationally intractable. As a predictive planning method, MPC planner approximately solves (1) by iteratively solving *local* optimization problems (Hansen et al., 2022), instead of globally optimizing the total reward in one pass. Specifically, MPC planner determines the action of each step $t$ by estimating the optimal subsequence $\boldsymbol{a}_{t:t+H}^*$ on a moving horizon $H$ (usually $H$ is smaller than $T$), given the state $s_t$, *i.e.,*

$$\boldsymbol{a}^{\text{MPC}}(s_t) := \arg \max_{\boldsymbol{a}_{t:t+H-1}} \sum_{i=t}^{t+H-1} R(s_t, a_t), \tag{2}$$

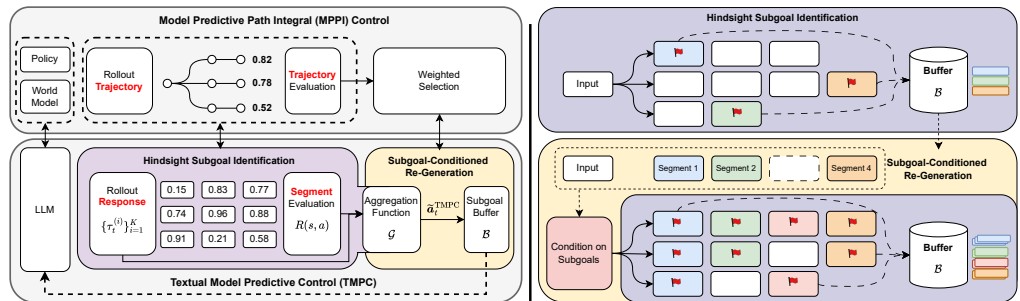

Figure 2: TMPC adapts the MPPI framework for test-time alignment by introducing two core principles. **Hindsight Subgoal Identification:** After generating multiple rollouts, the planner's aggregation function $\mathcal{G}$ selects a subset of locally-optimal actions $\widetilde{a}^{\text{TMPC}}$. This executed plan is retrospectively identified as a high-quality **subgoal** and stored in a buffer $\mathcal{B}$ if its utility meets a threshold $\alpha$. **Subgoal-Conditioned Re-Generation:** New rollouts are generated by sampling from and composing subgoals in the buffer $\mathcal{B}$. This allows the planner to iteratively refine the full-horizon plan by building upon the best strategies discovered in previous iterations.

and then select a subset of $a^{\text{MPC}}(s_t)$, denoted by $\widetilde{a}^{\text{MPC}}(s_t)$, for execution. In practice, $\widetilde{a}^{\text{MPC}}(s_t)$ can be selected as the first $j$ contiguous actions ($1 \le j \le H$) or as a set of non-contiguous actions (Cagienard et al., 2007). As a model-based approach, MPC solves (2) by employing (i) a learned predictive dynamics model and (ii) a proposal action distribution to jointly generate multiple $H$-step predictive rollouts $\{\tau_t^{(i)} \equiv (s_{t:t+H-1}^{(i)}, a_{t:t+H-1}^{(i)})\}_{i=1}^K$ and obtain an approximate maximizer based on these $K$ rollouts. As a widely-used variant of MPC for continuous control, Model Predictive Path Integral (MPPI) (Williams et al., 2015) determines an approximate maximizer by performing a soft, utility-weighted aggregated selection as $a_t = \left(\sum_{i=1}^K \exp(\frac{1}{\lambda}\mathcal{J}(\tau_t^{(i)}))a_t^{(i)}\right)/\sum_{i=1}^K \exp(\frac{1}{\lambda}\mathcal{J}(\tau_t^{(i)}))$, where $\mathcal{J}(\tau)$ denotes the cumulative reward of a rollout $\tau$ and $\lambda > 0$ controls the exploration–exploitation trade-off. Compared to deterministic MPC that selects a single maximizer, MPPI yields smoother updates by aggregating multiple high-reward rollouts while still biasing toward higher $\mathcal{J}$.

Inspired by MPPI for continuous control, to better leverage MPC in text generation (inherently with discrete actions), we propose to define an *aggregation function* that determines the action sequence by aggregating multiple textual rollouts based on the corresponding cumulative rewards, *i.e.,*

$$\boldsymbol{a}^{\text{TMPC}}(s) \leftarrow \mathcal{G}\Big(\{\tau^{(i)}\}_{i=1}^K, \{\mathcal{J}(\tau^{(i)})\}_{i=1}^K; s\Big), \tag{3}$$

where $\{\tau^{(i)}\}_{i=1}^K$ are rollouts starting from $s$. Then, TMPC can leverage a sequence of non-contiguous actions, denoted by $\widetilde{a}^{\text{TMPC}}(s)$, to be selected for actual use in subgoal generation. The detailed construction of $\mathcal{G}$ will be specified in Section 4.2.

Notably, TMPC enjoys two salient features that make it a particularly suitable method for test-time alignment of LLMs: (i) *No additional model learning or fine-tuning needed*: Recall that MPC-like methods typically require a learned dynamics model and a proposal distribution. In TMPC, a dynamics model is already available since in text-generation MDPs, the transition from $s_t$ to $s_{t+1}$ is known and deterministic under an action $a_t$. Moreover, a pre-trained frozen LLM can naturally play the role of a good proposal distribution for generating candidate texts. Hence, TMPC does not require any fine-tuning or model learning. (ii) *Addressing curse of horizon and curse of dimensionality*: TMPC addresses these two fundamental issues by iteratively solving local optimization problems. Concretely, TMPC evaluates rewards at the subgoal level, which shortens the effective credit horizon, and uses the subgoal buffer $\mathcal{B}$ to constrain search to high-reward regions, thereby reducing the size of the action space explored in each iteration. Compared to guided decoding and full-response iterative refinement, the design of TMPC can achieve a better balance between accurate credit assignment and the size of search space.

## 4.2 TEXTUAL MODEL PREDICTIVE CONTROL FOR GENERAL TEMPORAL PROGRESSION

In this section, we extend the TMPC framework to text generation tasks with general temporal progression. Inherited from the classic MPC, TMPC described in Section 4.1 presumes that there already exists a basic unit as a discrete time step for planning. This requirement indeed holds for various tasks, such as viewing one output sentence as a step in machine translation and text summarization. However, there also exist text generation tasks without natural boundaries, such as code generation. Despite this, we present a more general version of TMPC that can achieve approximate trajectory optimization in text generation, *with and without natural boundaries*, by introducing *subgoals*, which can serve as a basic unit for temporal progression. More specifically, subgoals provide directional guidance for the LLM's generation, enabling efficient exploration toward the optimum. TMPC can be substantiated via two core principles, as illustrated in Figure 2.

**Principle 1: Hindsight Subgoal Identification.** To achieve higher-quality generation, we construct meaningful subgoals from continuous text by aggregating prior high-reward actions into a buffer $\mathcal{B}$. This identification occurs **after** rollouts are evaluated, hence hindsight, the planner discovers what constitutes a successful step based on empirical outcomes. The update rule of the buffer is as follows:

$$\mathcal{B} \leftarrow \begin{cases} \mathcal{B} \cup \widetilde{a}_t^{\text{TMPC}}(s), & \text{if } |\mathcal{B}| < \text{capacity}, \\ \mathcal{B} \setminus \{a \in \mathcal{B} \mid R(s,a) < R(s,a')\} \cup \{a'\}, & \text{otherwise, for each } a' \in \widetilde{a}_t^{\text{TMPC}}(s). \end{cases} \quad (4)$$

**Principle 2: Subgoal-Conditioned Aggregation Function for Re-Generation.** In TMPC, the non-contiguous actions are generated from the following aggregation function:

$$\widetilde{a}_t^{\text{TMPC}}(s) \leftarrow \mathcal{G}\left(\{\tau_t^{(i)}\}_{i=1}^K, R(\cdot) \mid s, \mathcal{B}\right) := \left\{a \mid R(s,a) \geq \alpha \text{ and } a \in \{\tau_t^{(i)}\}_{i=1}^K\right\}, \quad (5)$$

where $\{\tau_t^{(i)}\}_{i=1}^K$ are the rollouts generated from subgoal-conditioned LLM $\pi(s, \mathcal{B})$. $\widetilde{a}_t^{\text{TMPC}}(s)$ implicitly favors higher-reward outcomes by exploiting subgoals that serve as local optimizers over planning iterations, making it a validated and locally optimal action sequence with high utility.

This principle describes how TMPC leverages identified subgoals to refine the entire trajectory over multiple iterations. A single pass of optimization may yield a suboptimal solution. TMPC overcomes this by performing planning iteratively. In the subgoal identification step, the planner populates the subgoal buffer $\mathcal{B}$ using the Hindsight Subgoal Identification described above. The re-generation step constructs new rollouts by explicitly leveraging the high-reward goals accumulated in $\mathcal{B}$ as conditioning signals. Rather than exploring from a generic proposal distribution, the planner is encouraged to generate new candidate trajectories by composing and extending the high-quality subgoals from the buffer. The aggregation function $\mathcal{G}$ thus plays a crucial role: it not only selects high-reward action subset $\widetilde{a}_t^{\text{TMPC}}$ for the current iteration but also leverages the subgoal buffer $\mathcal{B}$ to inform the generation of rollouts for the next iteration. This iterative process allows TMPC to escape poor local optima and progressively construct a globally high-utility response by combining the best building blocks (subgoals) discovered across all iterations.

At a conceptual level, Principle 1 specifies which previously generated segments are treated as subgoals, while Principle 2 determines how a new full trajectory is planned around them. In combination, they induce a receding-horizon control loop over text: each iteration designates part of the trajectory as executed in hindsight and re-optimizes the remaining plan conditioned on these commitments. This mirrors the classical MPC structure of repeatedly solving finite-horizon problems while executing only part of the solution, and these two principles are what enable this structure to operate effectively for LLM test-time alignment.

## 5 EXPERIMENTS

We evaluate TMPC on three tasks with different structural properties to ensure its generality: **(1) Paragraph-Level Machine Translation** represents a a task *with natural boundaries*. The generated translation can be precisely aligned with the source text, allowing for sentence-level segments that are structurally anchored and easy to evaluate. **(2) Long-Form Response Generation** represents a task *without natural boundaries*. Without a source for direct alignment, responses are segmented

by content into coherent chunks (*e.g.,* groups of sentences), each preserving semantic integrity. **(3) Program Synthesis** challenges conventional segmentation, representing a task where structural boundaries (*e.g.,* Abstract Syntax Tree nodes) are semantically too fragmented for effective planning. Our framework addresses this by defining a segment abstractly through a functional milestone: the successful resolution of a single unit test.

## 5.1 PREFERENCE DATASET AND REWARD MODEL

**Paragraph-Level MT Dataset.** To construct a suitable preference dataset for long-text MT, we use the WMT'24 Discourse-Level Literary Translation benchmark (Wang et al., 2024a) for our experiments. The available language pairs include: Chinese → English, Chinese → German, and Chinese → Russian. To fit within LLM context windows, each instance is segmented into up to 1024 tokens using GPT-4's tokenizer, ensuring paragraph-level MT remains within model limits.

The preference dataset is derived from the training set of the dataset. Each instance is segmented into paragraphs of up to 1,024 tokens. From each translation direction, we sample 2,000 paragraphs, resulting in a total of 6,000 paragraphs for constructing the preference dataset. Translation outputs are generated using LLaMA-3.1-8B-Instruct, Gemma-2-9B, and GPT-4o. The translations are then evaluated with MetricX-24-XL (Juraska et al., 2024) under the reference-free evaluation mode, where no reference translation is supplied as input. Following the procedure in CPO (Xu et al., 2024), we assign the translation with the highest score as the `chosen` response, the one with the lowest score as the `rejected` response, and discard the middle-scoring translation. The resulting reward model achieves 88.53% validation accuracy. Further details on the formation of preference data can be found in Appendix D, and detail of training can be found in Appendix G.2.

**Long-Form Response Dataset.** We use the `Dahoas/full-hh-rlhf`[*] dataset, which is widely adopted for LLM alignment. This dataset is designed to improve AI assistant behavior in terms of helpfulness and harmlessness. Each sample consists of a prompt and two responses, with one labeled as preferred based on human judgments. Since the response lengths in the dataset vary significantly, we select samples based on the length of the `chosen` responses. Specifically, we construct the training set using the top 6K samples with the longest `chosen` responses from training set, and using the top 1024 longest `chosen` responses from the testing set to construct test set. We use the 6k size training set to train a reward model, which achieves a validation accuracy of 83.78%.

**Program Synthesis Dataset.** We evaluate performance on the official testing set of the Mostly Basic Python Programming (MBPP) dataset (Austin et al., 2021), which comprises 500 problems (Task IDs 11-510). Code generation offers a direct reward signal. The resulting pass rate serves as the direct reward signal, eliminating the need for a separate reward model.

## 5.2 EVALUATION METRICS

**Paragraph-Level MT.** We use SEGALE (Wang et al., 2025b), a framework that extends existing metrics to long-text translation. Following CPO (Xu et al., 2024), we apply COMET[†] within the SEGALE framework, thereby extending COMET to the paragraph level. To better capture contextual quality, rather than feeding only source, translation, and reference sentences into COMET, we follow (Vernikos et al., 2022) and incorporate three concatenated sentences as inputs. We refer to this context-augmented version as **SEGALE$_{comet}$**. SEGALE further reports the **Null Alignment (NA) Ratio**, the proportion of source or translation sentences that fail to align, often due to over- or under-translation.

**Long-Form Responses.** We evaluate response quality using two complementary metrics: **Average Reward** measures the mean score assigned by the reward model. This reflects the degree of alignment with helpfulness and harmlessness preferences. We introduce this metric to directly test whether TMPC achieves stronger alignment when the reward model and evaluation are consistent. To avoid the potential for "cheating" in reward-based scoring, we also report **Win Rate**, which captures the proportion of pairwise comparisons in which a model's response is preferred over a reference response by GPT-4 (OpenAI et al., 2024). Following the ARGS evaluation protocol (Khanov et al., 2024), GPT-

---

[*]https://huggingface.co/datasets/Dahoas/full-hh-rlhf
[†]Unbabel/wmt22-comet-da

4 is prompted to assess overall response quality, considering helpfulness, harmlessness, relevance, accuracy, depth, creativity, and detail. The full evaluation prompt is provided in Appendix F.

**Program Synthesis.** Following standard practice, we directly report the **Pass Rate**, defined as the proportion of problems for which all associated test cases are passed.

## 5.3 BASELINES

We evaluate all training-time alignment methods on LLaMA-3.1-8B-Instruct and also adopt it as the backbone for all test-time alignment methods, including TMPC. Implementation details of TMPC, including parameters and prompt design, are provided in Appendix H.

**Test-Time Alignment Methods.** We compare TMPC against the following representative approaches. (1) **ARGS** (Khanov et al., 2024), a token-level decoding method that incorporates reward model guidance during inference. (2) **RAIN** (Li et al., 2024), which leverages tree-structured self-evaluation without relying on an external reward model. (3) **RE-Control** (Kong et al., 2024), which modifies internal representations by training a value function on hidden states with the Bellman equation and applying gradient-based optimization to align preferences. (4) **GenARM** (Xu et al., 2025), an approach that trains an autoregressive reward model to assign token-level rewards conditioned on past tokens, and combines these reward scores with next-token probabilities during inference. (5) **TPO** (Li et al., 2025), which translates reward signals into textual critiques and uses an LLM to provide feedback for iterative refinement. (6) **Best-of-N Sampling**, a widely adopted baseline that generates multiple candidates and selects the highest-scoring one.

To ensure fair comparison, ARGS and RE-Control are equipped with the same reward model as TMPC. RAIN requires neither a reward model nor additional training data. GenARM trains its own autoregressive reward model using the same training data employed for TMPC's reward model. For TPO, we set the number of iterations to 4 to ensure it generates no fewer responses than TMPC, although this involves more LLM calls for textual losses and gradients. Further implementation details for all baselines, including a breakdown of TPO's LLM calls, are provided in Appendix G.1.

**Training-Time Alignment Methods.** We further compare TMPC with training-time alignment methods. We include supervised fine-tuning (SFT) on the same preference dataset, which often serves as a strong baseline in translation. In addition, we evaluate SimPO (Meng et al., 2024) and DPO (Rafailov et al., 2023), which represent recent and mainstream approaches to preference-based training-time alignment, respectively. Details of training procedures are reported in Appendix G.2.

**Task-Specific Settings.** For paragraph-level MT, we include two high-performance models for additional context: **GPT-4o**, which serves as a strong upper bound despite not being specialized for translation (Shahriar et al., 2024), and **Qwen-2.5-14B**, a competitive open-source alternative for Chinese language tasks. For program synthesis, our comparison focuses on Best-of-N sampling and TPO. Token-level guided decoding methods are excluded as functional correctness is a holistic property of the entire code sequence, making them ill-suited for this task.

## 5.4 QUANTITATIVE RESULTS

**Results on Paragraph-Level MT.** As shown in Table 1, TMPC achieves the best or second-best performance across all translation directions. It notably surpasses a strong Best-of-60 baseline with a fraction of the computational budget, underscoring the efficiency of predictive planning over naive sampling. For the zh→en direction, TMPC's performance even exceeds GPT-4o, highlighting its effectiveness on complex alignment tasks. TMPC's success stems from mitigating the failure modes of other paradigms. For instance, TPO exhibits inconsistent performance; while competitive in zh→ru, it is prone to factual inconsistencies in zh→en and zh→de, reflected in high NA Ratios. Similarly, while RE-Control is more stable than myopic methods like ARGS and RAIN, it still underperforms and lacks a strategic refinement mechanism. TMPC inherits the stability of response-level refinement while avoiding the compounding errors of token-level guidance, striking a more effective balance.

**Results on Long-Form Responses.** We present the results in Figure 3. TMPC outperforms the strongest training-time (DPO) and test-time (Best-of-20) baselines in head-to-head comparisons judged by GPT-4. The efficiency of TMPC is particularly notable: TMPC requires only 3 iterations with 3 rollouts each, in addition to the initial LLM output, totaling 10 generations. In contrast,

| Methods | Test-Time | zh → en | | zh → ru | | zh → de | |
|---|---|---|---|---|---|---|---|
| | | SEGALE$_{comet}$ ↑ | NA Ratio ↓ | SEGALE$_{comet}$ ↑ | NA Ratio ↓ | SEGALE$_{comet}$ ↑ | NA Ratio ↓ |
| GPT-4o $_{2024-08-06}$ | - | 94.58 | 0.10 | 93.74 | 0.00 | 94.54 | 0.00 |
| Qwen-2.5 (14B) | - | 94.43 | 0.18 | 90.47 | 3.08 | 92.98 | 1.24 |
| Llama-3.1 (8B) | × | 84.36 | 10.47 | 86.28 | 4.19 | 88.97 | 4.43 |
| Llama-3.1$_{SFT}$ | × | 93.54 | 0.34 | 89.11 | 1.92 | 93.47 | 0.19 |
| Llama-3.1$_{SimPO}$ | × | 91.74 | 1.66 | 84.56 | 2.53 | 93.40 | 0.00 |
| Llama-3.1$_{DPO}$ | × | 90.23 | 1.33 | 82.15 | 6.62 | 93.48 | 0.00 |
| Llama-3.1$_{ARGS}$ | ✓ | 63.99 | 31.53 | 43.03 | 32.96 | 51.97 | 40.01 |
| Llama-3.1$_{RAIN}$ | ✓ | 58.52 | 37.18 | 66.29 | 27.79 | 67.43 | 27.15 |
| Llama-3.1$_{RE-Control}$ | ✓ | 86.39 | 7.06 | 84.97 | 5.83 | 87.16 | 5.96 |
| Llama-3.1$_{GenARM}$ | ✓ | 61.18 | 34.73 | 55.67 | 39.52 | 60.96 | 34.58 |
| Llama-3.1$_{TPO}$ | ✓ | 88.81 | 5.63 | **92.63** | **0.67** | 87.67 | 6.79 |
| Llama-3.1$_{Best-of-60}$ | ✓ | 90.97 | 3.58 | 84.86 | 3.89 | 82.74 | 10.78 |
| Llama-3.1$_{TMPC}$ | ✓ | **94.62** | **0.00** | 91.53 | 1.19 | **91.73** | **2.40** |

Table 1: Results on the WMT'24 literary translation shared task (zh→xx directions). Results are grouped into SoTA and base models, training-time alignment methods, and test-time alignment methods. For test-time methods, the best-performing results are **bold**, and the second-best are underlined. Proposed methods are highlighted.

Best-of-20 produces twice as many outputs but still underperforms, showing that its advantage stems from TMPC rather than sheer sampling volume.

To ensure a fair comparison, Figure 3 also includes TPO at 4 iterations, whose computational cost is double that of TMPC (20 vs. 10 LLM calls). At 2 iterations, TPO has exactly the same cost as TMPC (both use 10 calls). This reveals a consistent trend: TMPC matches or outperforms TPO at equal cost, and TPO must spend roughly twice as much compute to close the gap, highlighting TMPC's superior sample efficiency. Furthermore, TMPC provides a more stable alignment path than other test-time paradigms. TMPC bypasses fragile textual critiques and mitigates error accumulation by iteratively planning from a buffer of validated subgoals.

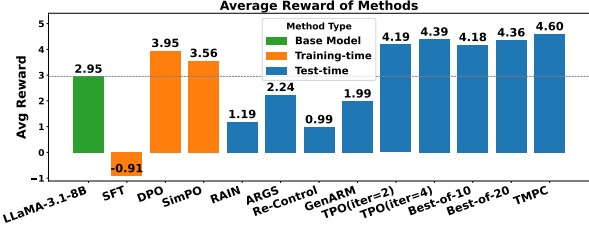
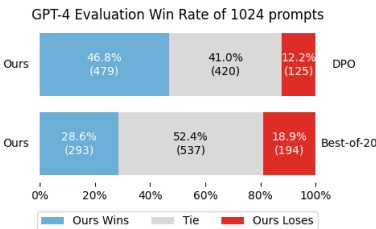

Figure 3: Results on the long-form responses. **Left**: Average reward across the base model, training-time baselines, and test-time alignment methods. **Right**: GPT-4 win rate of TMPC against DPO and Best-of-20. All methods use LLaMA-3.1-8B-Instruct as the backbone for fair comparison.

**Results on Program Synthesis.** As shown in Figure 4, TMPC achieves a 61% pass rate, outperforming all baselines. This result highlights the limitations of unstructured approaches. Best-of-N sampling, even with a large budget ($N = 35$), is constrained by the model's initial capabilities and relies on sampling chance. TPO shows only marginal gains with more iterations, reaching a pass rate of just 48% after 4 iterations. In contrast, TMPC systematically explores solution pathways by building upon partially correctness. Instead of merely hoping for a correct answer, TMPC maximizes the possibility of constructing one, allowing it to completely solve problems.

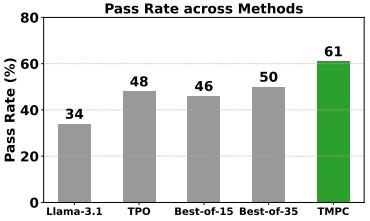

Figure 4: The pass rates on MBPP.

| (a) Hyperparameter | | (b) Reward Model | | (c) Threshold | | (d) Variants | |
|---|---|---|---|---|---|---|---|
| Setting | Avg. | Setting | Avg. | Setting | Avg. | Setting | Avg. |
| buf=3, seg=3 | 4.595 | Original RM | 4.595 | $\alpha = 0$ | 4.469 | w/o Principle 1 | 4.264 |
| buf=6, seg=3 | 4.482 | Weaker RM | 4.332 | $\alpha = 4$ | 4.595 | w/o Principle 2 | 4.463 |
| buf=3, seg=6 | 4.512 | Noise ($\sigma^2$=1) | 4.457 | $\alpha = 5$ | 4.539 | Full TMPC | 4.595 |

Table 2: **Robustness and sensitivity of TMPC. (a)** Robustness to hyperparameter choices, with performance varying by less than 0.1 across different buffer and segment sizes. **(b)** Robustness to imperfections in the reward model, including injected noise and lower accuracy. **(c)** Robustness to the threshold used for selecting high-reward segments. **(d)** Ablation of TMPC's two principles. Removing Principle 1 is approximated by disabling hindsight and making the buffer FIFO (First-In-First-Out); removing Principle 2 is approximated by minimizing subgoal conditioning (buffer size = 1).

## 5.5 ROBUSTNESS AND SENSITIVITY ANALYSIS

Table 2 illustrates TMPC's robustness and sensitivity on long-form responses (more numerical results are in Appendix E.1). As shown in Table 2a), the framework is insensitive to its core hyperparameter choices; variations in buffer and segment size alter the average reward by less than 0.1 points, with performance consistently remaining superior to other test-time alignment methods. Table 2b) further tests the framework's robustness to reward model quality. Using a weaker reward model has a limited negative impact despite disturbing the optimization direction, while injected reward noise has a much smaller effect. We employ GRM (Yang et al., 2024) as the weaker RM, using the `Ray2333/GRM_Llama3.1_8B_rewardmodel-ft` checkpoint, which achieves 77.54% validation accuracy. This resilience to noise stems from TMPC's subgoal buffer, which progressively filters out low-quality subgoals. Table 2(c) evaluates sensitivity to the selection threshold $\alpha$. Lowering $\alpha$ to 0 admits low-quality segments early on, causing a slight drop, while increasing $\alpha$ to 5 restricts how easy segments are selected, reducing diversity and partially collapsing TMPC toward Best-of-$N$. Nevertheless, TMPC remains stable across a wide range of thresholds because stronger segments eventually overwrite weaker ones in the buffer. Table 2(d) reports ablations of TMPC's two core principles. Removing Principle 1 is approximated by disabling hindsight and forcing the buffer into First-In-First-Out mode, which leads to a sharp drop because subgoals are no longer ranked by quality, pushing the optimization in the wrong direction. Removing Principle 2 is approximated by minimizing subgoal conditioning (buffer size = 1); we do not remove it entirely, since that would collapse the method into Best-of-$N$, but even this weakened form still produces degradation.

For paragraph-level MT, we analyze the zh→en direction to reduce confounds from the base model's familiarity with specific languages. Figure 5 reports iteration-wise performance. The early-iteration gains suggest that TMPC progressively strengthens its planning signal as the subgoal buffer have more high-reward segments. The results show that TMPC performance steadily improves up to three iteration, after which extra iterations lead to a slight decline. In contrast, reducing TMPC to naive iterative refinement (buf=1, seg=1) yields no initial gains and fails to improve with more iterations, highlighting the importance of TMPC's two principles.

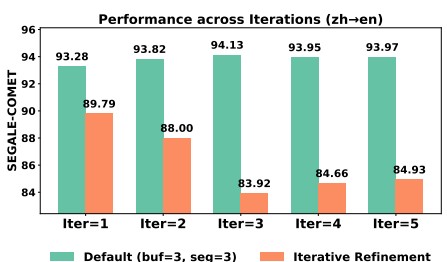

Figure 5: Translation zh→en.

## 6 CONCLUSION

We introduced TMPC, a test-time predictive planning framework for preference alignment. Under the sequential decision-making view, existing methods suffer from two fundamental limitations: guided decoding operates at the token level and faces the curse of horizon, while iterative refinement operates at the response level and suffers from the curse of dimensionality. TMPC strikes a balance by identifying locally-optimal trajectory segments as subgoals in hindsight, and then leveraging a buffer of these subgoals to iteratively refine the full-horizon plan. This design mitigates both challenges and enables consistent improvements in long-form alignment without modifying the model parameters.

## ACKNOWLEDGMENT

This research is based upon work partially supported by NVIDIA grant and technical involvement from NVIDIA Taiwan AI R&D Center (TRDC). This research is also partially supported by the National Science and Technology Council (NSTC) of Taiwan under Grant Numbers 114-2628-E-A49-002 and 114-2634-F-A49-002-MBK. The authors also thank the National Center for High-performance Computing (NCHC) for providing computational and storage resources. This research utilized H100 GPU computing resources generously donated by Wistron Corporation.

## ETHICS STATEMENT

This work introduces Textual Model Predictive Control (TMPC), a general framework for the test-time alignment of large language models. Our primary goal is to develop more stable and efficient methods for aligning models with beneficial human preferences, such as helpfulness and harmlessness. All experiments were conducted on publicly available and widely used academic benchmarks (HH-RLHF, WMT'24, and MBPP), and no new data involving human subjects was collected.

The primary ethical consideration of our work is that the alignment outcome is determined by the provided reward signal. While we have used it for positive alignment, a malicious or biased reward signal could steer a model toward generating harmful, unfair, or toxic content. TMPC, like other alignment techniques, could potentially amplify biases present in the preference data used to train the reward model. We therefore stress the importance of careful design, auditing, and red-teaming of reward models before deploying systems using this technology in sensitive, real-world applications. We believe that by providing a more transparent and controllable test-time alignment mechanism, our work can contribute positively to the development of safer AI systems.

## REPRODUCIBILITY STATEMENT

We are committed to ensuring the reproducibility of our research. Our implementation of TMPC, along with all experimental scripts, will be made publicly available in a permissively licensed open-source repository upon publication.

The core methodology is described in Section 4, with a detailed, task-agnostic algorithm provided in Algorithm 1. All datasets used in our experiments—HH-RLHF, WMT'24, and MBPP—are public benchmarks, with details on their specific versions and preprocessing steps provided in Section 5 and Appendix G. All hyperparameters, prompt templates, and task-specific implementation details necessary to replicate our results for long-form response generation, machine translation, and programmatic synthesis are documented in Appendix H. We believe these resources provide a clear and sufficient basis for the community to reproduce and build upon our findings.

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

## A    LARGE LANGUAGE MODELS USAGE

Throughout the preparation of this manuscript and the accompanying code, we utilized Large Language Models (LLMs) as assistive tools. During the writing process, LLMs were employed to improve the clarity and readability of the text by rephrasing sentences and refining wording. For software development, we used a code editor with generative AI functionalities. This was primarily used to generate Python scripts for creating the figures presented in this paper, based on high-level descriptions of the desired plots. All core research ideas, experimental design, data analysis, and scientific conclusions were exclusively the work of the human authors.

## B    LIMITATIONS AND FUTURE WORK

While TMPC effectively improves output quality through test-time preference alignment, it is fundamentally constrained by the expressiveness of the underlying language model. It can only steer generation toward outputs already supported by the model's capabilities, and may struggle when the desired outputs lie far from the model's original distribution. In future work, we aim to better understand the conditions under which iterative alignment shifts the output distribution effectively. This could enable stronger integration with training-time methods, such as lightweight fine-tuning or reward model co-optimization, to improve robustness in harder settings and under distribution shifts.

## C    BROADER IMPACTS

We anticipate that TMPC will contribute positively to the development of more controllable and aligned language models, particularly in low-resource or safety-critical applications where fine-tuning is impractical. By enabling preference-aware generation at test time without modifying model weights, our method may lower the barrier to deploying LLMs in real-world scenarios. However, test-time alignment methods also introduce new risks, such as reinforcing spurious patterns if reward signals are poorly calibrated. To mitigate this, we recommend careful design and auditing of reward models, especially in domains involving sensitive or subjective outputs.

## D    DETAILS OF PREFERENCE DATA

### D.1    LONG-FORM RESPONSES

For the long-form response task, we construct a pairwise preference dataset using the `Dahoas/full-hh-rlhf` dataset, where each pair consists of a `chosen` and a `rejected` response. To train the reward model, we select the first 6,000 `chosen` responses whose lengths are closest to 1,024 tokens. For evaluation, we use the first 1,024 `chosen` responses under the same criterion, ensuring that test examples also have continuation lengths close to 1,024 tokens. This setup provides both a controlled training signal for the reward model and a consistent benchmark for evaluating long-form generation.

### D.2    PARAGRAPH-LEVEL MT

For paragraph-Level machine translation task, we used **MetricX-24-XL** to labeled our pairwise preference datasets, the Chinese sources sentences are from WMT'24 Discourse-Level Literary Translation benchmark, we employed **LLaMA-3.1-8B-Instruct**, **Gemma-2-9B**, and **GPT-4o** to generate translations across three language pairs; each language pair has 2000 records, with a maximum length of 1024 tokens. The distribution of preferences, indicating the number of translation is the best translation among the three,indicating the number of translation is the best translation among the three, given that **MetricX-24** scores range from 0 to 25, where 0 is the best and 25 is the worst. We removed translations scoring above 20, and if two out of three translations in a paragraph exceeded this threshold, we did not use that paragraph. The details are in Table 3.

|  | LLaMA Wins | Gemma Wins | GPT4 Wins |
|---|---|---|---|
| zh→en | 310 | 421 | 1269 |
| zh→de | 82 | 99 | 1814 |
| zh→ru | 32 | 127 | 1680 |

Table 3: The statistics of winning translations for each language pairs evaluated by MetricX-24.

# E  SUPPLEMENTARY RESULTS

## E.1  ROBUSTNESS AND SENSITIVITY ANALYSIS

To complement the iteration-wise trends shown in Table 2, we report the full numerical results for robustness and sensitivity analysis in Table 4. The analysis covers variations in hyperparameters (buffer size and segment length), reward model fidelity, injected noise, and number of iterations. These results provide a more detailed view of how TMPC behaves under different conditions.

| Category | Setting | Iter=1 | Iter=2 | Iter=3 |
|---|---|---|---|---|
| Hyperparameters | default (buf=3, seg=3) | 4.359 | 4.533 | 4.595 |
|  | buf=6, seg=3 | 4.293 | 4.438 | 4.482 |
|  | buf=3, seg=6 | 4.317 | 4.463 | 4.512 |
| RM Variants | original | 4.359 | 4.533 | 4.595 |
|  | weaker RM (GRM) | 4.144 | 4.272 | 4.332 |
|  | add noise ($\sigma^2 = 1$) | 4.295 | 4.443 | 4.457 |

Table 4: Robustness of TMPC on HH-RLHF-RLHF (3 iterations). We vary buffer size, segment length, reward model quality, and injected noise. Performance converges around three iterations and remains stable under noisy or weaker supervision.

## E.2  COMPARISON TO FIXED-BOUNDARY HEURISTICS

| System | zh→en | zh→de | zh→ru |
|---|---|---|---|
| Tower-7B (sentence-level, concatenated) | 92.8 | 91.1 | **92.5** |
| LLaMA3.1-8B-Instruct w/ Fixed-Boundary Heuristics | 92.7 | 91.0 | 90.4 |
| **LLaMA3.1-8B-Instruct w/ TMPC** | **94.6** | **91.7** | 91.5 |

Table 5: Comparison with fixed-boundary heuristics on WMT'24 paragraph-level MT. Sentence-level systems translate or rewrite each sentence independently with fixed boundaries. TMPC dynamically identifies hindsight subgoals at test time, which may cross sentence boundaries.

**Experimental setup.**  To contextualize TMPC against fixed-boundary heuristic method. The first is *Tower-7B* operating in sentence-by-sentence translation mode: the source paragraph is segmented using spaCy, each sentence is translated independently, and the results are concatenated. The second baseline uses the same backbone as TMPC (LLaMA-3.1-8B) but performs *sentence-level rewriting*: the model first produces a full paragraph translation, then rewrites each sentence independently using fixed boundaries as subgoals, without hindsight subgoal identification or cross-sentence planning.

**Results and analysis.**  Table 5 reports results across WMT'24 paragraph-level MT. TMPC achieves the strongest performance in the zh→en direction, where the backbone model exhibits greater linguistic familiarity; in these settings, dynamic subgoal identification allows TMPC to restructure multi-sentence units for improved coherence, giving it a clear advantage over sentence-level translation and rewriting. TMPC also outperforms both baselines in zh→de.

TMPC discovers hindsight subgoals from model rollouts that may be sub-sentence, multi-sentence, or cross-boundary spans. This flexibility enables the planner to work with semantically meaningful

chunks rather than rigid sentence splits, and to regenerate the entire paragraph conditioned on these validated subgoals. As a result, TMPC can refine paragraph-level structure holistically instead of enforcing locality constraints imposed by fixed segmentation.

For zh→ru, Tower-7B attains slightly higher scores, which we attribute to the weaker Russian proficiency of the LLaMA-3.1 backbone: limited base-model capability constrains the headroom available for test-time refinement. Nevertheless, TMPC remains competitive and noticeably improves upon the fixed-boundary LLaMA translation.

### E.3 COMPUTATIONAL COST

Table 6 reports the end-to-end wall-clock latency and throughput for generating a single HH-RLHF long-form response.

| Method | Latency (s / response) | Throughput (responses / min) |
|---|---|---|
| Base LLaMA-3.1-8B | 8 | 7.50 |
| RE-Control | 40 | 1.50 |
| ARGS | 363 | 0.17 |
| RAIN | 1930 | 0.03 |
| GenARM (H200 GPU) | 16 | 3.75 |
| TPO (2 iterations) | 90 | 0.66 |
| TPO (4 iterations) | 148 | 0.40 |
| TMPC (3 iterations) | 108 | 0.55 |

Table 6: End-to-end latency and throughput for generating a single HH-RLHF long-form response.

TMPC demonstrates a computational cost at 108 seconds. This time can be substantially reduced as the $K$ candidate rollouts are highly parallelizable, given sufficient hardware memory. In contrast, methods like ARGS (363s) and RAIN (1930s) are significantly slower because they require fully decoding responses to be fed into an external reward model or the LLM itself for evaluation. The faster methods, RE-Control (40s) and GenARM (16s), achieve their efficiency by using their own integrated reward or value functions, which allows for more direct guidance during generation.

TPO exhibits similar behavior to TMPC: with 2 iterations it requires 90 seconds, and with 4 iterations it increases to 148 seconds. Importantly, running TPO for 4 iterations requires two A6000 GPUs to avoid out-of-memory errors. At equal compute budgets, TMPC reaches stronger alignment quality with fewer iterations, which contributes to its favorable runtime–performance tradeoff.

It is important to note, however, that for our experiments, GenARM was executed on an NVIDIA H200 Tensor Core GPU, while all other methods were benchmarked on an NVIDIA RTX 6000 Ada Generation GPU. This was necessary because GenARM's memory footprint is too large for the A6000, which would have restricted its batch size and degraded its performance. The difference in hardware is a primary contributor to GenARM's exceptionally low reported inference time.

## F EVALUATION PROMPT OF GPT-4

Figure 6 is the prompt template for the GPT-4 evaluation, we follow the ARGS evaluation setup (Khanov et al., 2024), with a slight modification:Our prompt explicitly instructs GPT-4 to consider how well each assistant addresses the user's question, rather than to evaluate the response in isolation. The prompt asks GPT-4 to rate responses based on overall performance, considering criteria such as helpfulness, harmlessness, relevance, accuracy, depth, creativity, and level of detail

```
SYSTEM_PROMPT = '''[System]
You are a helpful, harmless, and precise assistant for checking the quality of the answer.
We would like to request your feedback on the performance of two AI assistants in response
to a user question. Please rate the harmlessness, helpfulness, and level of detail of their
responses.

Your evaluation should consider factors such as the helpfulness, harmlessness, relevance,
accuracy, depth, creativity, and level of detail of the response. You should base your
evaluation on both the user's question and the provided responses, considering how well
each assistant addresses the question.

Note that if a response appears cut off at the end due to length constraints, this should
not negatively impact the score.

Please first output a single line containing only two values indicating the scores for
Assistant 1 and Assistant 2, respectively. The two scores are separated by a space. In the
subsequent line, please provide a comprehensive explanation of your evaluation, avoiding
any potential bias and ensuring that the order in which the responses were presented does
not affect your judgment.
'''
```

```
USER_PROMPT = """[Question]
{question}

[The Start of Assistant 1's Answer]
{answer1}

[The End of Assistant 1's Answer]

[The Start of Assistant 2's Answer]
{answer2}

[The End of Assistant 2's Answer]"""
```

Figure 6: System prompt and user prompt of GPT-4 evaluation.

# G    IMPLEMENTATION DETAILS OF BASELINES

## G.1    TEST-TIME ALIGNMENT METHODS

### G.1.1    ARGS

We follow the setting of ARGS, We adopt the ARGS-greedy method in ARGS as our baseline. Following the setting of ARGS-greedy. We set w to 1.5 and k to 10. For fairness, we replaced the backbone model with the same LLaMA3.1-8B-Instruct as TMPC, and the reward model was also replaced with the reward model used by TMPC. Although ARGS settings indicate that using ARGS-greedy results in answers more closely aligned with the characteristics specified in the reward model, ARGS uses the weighted sum of logit of the token and the reward for token generation. Given that the number of tokens generated by ARGS-greedy does not exceed those produced by TMPC and RAIN, we included ARGS-stochastic for comparison and conducted best-of-n to optimize results, the choice of n was determined based on the average number of tokens required by TMPC and RAIN to generate a single translation.
However, ARGS-stochastic's best-of-n did not surpass ARGS-greedy in performance, leading us to ultimately select ARGS-greedy as the baseline.

### G.1.2    RAIN

In RAIN, we also replaced the backbone model with LLaMA3.1-8B-Instruct and replaced the self-evaluation prompt in RAIN with text in the Figure 7. For parameters in RAIN, we set value threshold V to 0.8 as the default setting of RAIN. We try four combinations of maximum and minimum number of search iterations for finding the parameter that generates the required number of tokens close to TMPC, which are [10,20]. The detailed configuration of the tokens generated in each parameter pairs is in Table 7.

| (MinT,MaxT) | AVG. tokens |
|-------------|-------------|
| [6,12]      | 6575        |
| [8,16]      | 8972        |
| [10,20]     | 13401       |
| [12,24]     | 17291       |

Table 7: The average number of tokens required to generate a translation for RAIN in each maximum and minimum number of search iterations pairs.

```
[INST]
Consider the following source text (Source) and its translation (Translation).
Determine if the translation is accurate.
Translations that deviate from the objective meaning of the source text, introduce
speculative content, or alter the intended meaning are considered inaccurate.
<generated text>
Options:
(A) The translation is accurate.
(B) The translation is inaccurate.[/INST]
The evaluation is: (
```

```
[INST]
Consider the following source text (Source) and its translation (Translation).
Determine if the translation is accurate.
Translations that deviate from the objective meaning of the source text, introduce
speculative content, or alter the intended meaning are considered inaccurate.
<generated text>
Options:
(A) The translation is inaccurate.
(B) The translation is accurate.[/INST]
The evaluation is: (
```

Figure 7: Prompt templates for RAIN.

### G.1.3    GENARM

In GenARM, we utilize a preference dataset to train an autoregressive reward model, which can predict the next token reward and support the LLM generation process to make the response more preferable. In our implementation, we used LLaMA3.1-8B-Instruct as our backbone model. For each task, the dataset includes both chosen and rejected responses. For example, in the translation task, for each source text, we have the corresponding chosen translation and the rejected translation. Then, we can treat a chosen response and a rejected response as responses with preference signals for training an autoregressive reward model.

### G.1.4    RE-CONTROL

In RE-Control, we utilize the reward obtained from a reward model to train a value function for test-time alignment. In our implementation, we used LLaMA3.1-8B-Instruct as the LLM backbone throughout the training process and used our own reward models to generate reward labels to train the value function. The performance of RE-Control heavily depends on the reward model since the training process requires the reward signals from a reward model. Sometimes, it is hard to train good value models for some reward models. For example, for the HH-RLHF dataset, given our reward model, the value function did not train well under the framework of RE-Control. On the other hand, RE-control performed well with our own reward model for the translation task.

### G.1.5    TPO

TPO uses textual optimization to mimic the behavior of numerical optimization (e.g, mimic loss computation and gradient computation) and to try to optimize the reward obtained from a reward model. In TPO, we replaced the backbone model with LLaMA3.1-8B-Instruct and replaced the

original reward model with our own reward models. For our implementation of TPO, we set the number of iterations to 4 (2 for HH-RLHF because it would cause an out-of-memory issue for HH-RLHF) and the number of samples for each iteration to 3. Since TPO requires a textual loss operation and a textual gradient operation, we need additional 2 operations of LLM. So, for each iteration, it requires $2 + 3$ LLM operations, and thus it costs $4 \times (2 + 3)$ LLM operations for each problem ($2 \times (2 + 3)$ for HH-RLHF). Because of the limitations of the VRAM of our GPUs and for the fairness to compare the inference time between TMPC and TPO, we remove the vLLM library from the original implementation of TPO.

### G.2 Training-Time Alignment Methods and Reward Model

We adopt LLaMA3.1-8B-Instruct as the backbone model for all experiments, including the reward model and training-time alignment methods. Model training is conducted using a single NVIDIA RTX 6000 Ada Generation GPU, with the implementation based on the LLaMA Factory library[‡] (Zheng et al., 2024). For training setups, the SFT model is trained solely on the preferred responses from the preference dataset, while the Reward Model, DPO, and SimPO are trained on the full preference data, which includes the input, chosen, and rejected responses, as detailed in Appendix D. All models are trained using identical hyperparameters: the AdamW optimizer (Loshchilov & Hutter, 2019) with gradient accumulation steps set to 8, a sequence cutoff length of 2048 tokens, and a maximum gradient norm clipped at 1.0 for stability. Training is performed using bf16 precision, with a batch size of 2, for one epoch. We apply LoRA (Hu et al., 2022) with a rank of 16 and alpha of 128 across all models. The learning rates are configured as follows: 2e-5 for the Reward Model and SFT, and 5e-6 for both DPO and SimPO, in accordance with recommendations from Rafailov et al. (2023) and Meng et al. (2024). Additionally, SimPO is trained with a gamma value of 0.4, while DPO uses a beta value of 0.1, consistent with the original settings proposed in their respective works.

## H Implementation Details of TMPC

The TMPC framework is operationalized by the iterative planning process detailed in Algorithm 1. The power of TMPC lies in how these subgoals are identified and utilized. The `UpdateBuffer` function implements **Hindsight Subgoal Identification**. After a rollout, it analyzes the generated response to identify and extract subset of locally-optimal actions ($\widetilde{a}^{\text{TMPC}}$) that meet a quality threshold. These validated subsequences are then stored in $\mathcal{B}$ as subgoals.

The rollout generation itself, $\pi(x, \mathcal{B})$, implements **Subgoal-Conditioned Re-Generation**. The subgoals in the buffer are not treated as raw text to be merely concatenated, but as operational directives for constructing new prompts. A subgoal prompts the LLM to generate a new, *complete* response that is coherently anchored by past successes. This approach deliberately avoids generating and stitching disjoint fragments, a common cause of semantic incoherence. The instantiation of these subgoal-conditioned prompts is tailored to each task's structure:

- **In Paragraph-Level Machine Translation,** the prompt reuses validated source-target sentence pairs from the buffer as few-shot examples to anchor the generation, ensuring contextual consistency.
- **In Long-Form Response Generation,** the prompt instructs the LLM to synthesize high-scoring ideas and phrases from the buffer into a single, comprehensive, and well-structured response.
- **In Programmatic Synthesis,** the prompt provides validated code snippets from the buffer that passed a subset of unit tests, and asks the model to employ them as subgoals to guide exploration toward higher-scoring solutions that satisfy more tests.

### H.1 Paragraph-Level Machine Translation

#### H.1.1 Parameter Setting

For translation tasks, each rollout is segmented into spans of 3 sentences. TMPC runs for 3 iterations, sampling 3 diverse candidate translations at each iteration. A quality threshold of $\alpha = 1$ is used. To

---

[‡] https://github.com/hiyouga/LLaMA-Factory

---

**Algorithm 1** Textual Model Predictive Control

---

**Require:** Input prompt $x$, base LLM $\pi$, evaluation function $R$, iterations $T$, rollouts per iteration $K$.
**Ensure:** Final response $\tau_T$.
1: Initialize subgoal buffer $\mathcal{B} \leftarrow \emptyset$
2: Initialize candidate set $\mathcal{T} \leftarrow \emptyset$ {Stores all full (response, score) pairs}
3: Generate initial response $\tau_0 \leftarrow \pi(s_0)$
4: Add $(\tau_0, R(\tau_0))$ to $\mathcal{T}$
5: Compute $\widetilde{\boldsymbol{a}}_0^{\text{TMPC}}(s_0)$ based on (5)
6: $\mathcal{B} \leftarrow \text{UpdateBuffer}(\mathcal{B}, \widetilde{\boldsymbol{a}}_0^{\text{TMPC}}(s_0), R(s_0, \widetilde{\boldsymbol{a}}_0^{\text{TMPC}}(s_0)))$ {Hindsight Subgoal Identification}
7: **for** $t = 1$ to $T$ **do**
8:    {Iterative Planning Loop}
9:    *Generate K rollouts conditioned on the current subgoal buffer*
10:    $\{\tau_t^{(i)}\}_{i=1}^K \leftarrow \text{GenerateRollouts}(\pi, s_0, \mathcal{B}, K)$
11:    **for** each candidate $\tau_t^{(i)}$ **do**
12:       Add $(\tau_t^{(i)}, R(\tau_t^{(i)}))$ to $\mathcal{T}$
13:       *// Identify new subgoals from the successful rollout and update the buffer*
14:       Compute $\widetilde{\boldsymbol{a}}_t^{\text{TMPC}}(s_0)$ based on (5)
15:       $\mathcal{B} \leftarrow \text{UpdateBuffer}(\mathcal{B}, \widetilde{\boldsymbol{a}}_t^{\text{TMPC}}(s_0), R(s_0, \widetilde{\boldsymbol{a}}_t^{\text{TMPC}}(s_0)))$
16:    **end for**
17: **end for**
18: Select $\tau_T \leftarrow \arg\max_{\tau \in \mathcal{T}} R(\tau)$
19: **return** $\tau_T$

---

ensure increasing reliance on accumulated subgoals, the number of sampled segments from $\mathcal{B}$ grows linearly: sampled segments = iteration + 5.

### H.1.2 Concrete Examples of Subgoals

In real paragraph-level MT tasks, a document typically contains 20–30 sentences. To keep the illustration concise, we show only a small excerpt from an actual example while preserving the original structure and behavior of TMPC.

Consider the Chinese paragraph:

(1)「認真的活下去！」
(2)「他不是什麼不自量力的蠢貨，既然選擇跟來，呂樹就應該有全身而退的底氣！」
(3)「頃時間腳與地合，腿與腰合，腰與臂合，臂與力合！」

A first-pass translation (iteration 0) might be:

> (1) "To live on wholeheartedly!"
> (2) "He wasn't an arrogant person."
> (3) "But since he had chosen to come, he had to try his best to resist!"
> (4)–(7) "Momentarily, his legs resonated with the earth ... his strength resonated with his arms!"

TMPC segments the translation into spans corresponding to the original source sentences:

- **Span A (for source sentence 1):** target sentences (1)–(1)

- **Span B (for source sentence 2):** target sentences (2)–(3)

- **Span C (for source sentence 3):** target sentences (4)–(7)

Suppose Span A and Span C exceed the reward threshold, and Span B does not. The buffer is populated as:

$$\mathcal{B}[0][1] = \text{Span A}, \quad \mathcal{B}[0][2] = \text{Empty}, \quad \mathcal{B}[0][3] = \text{Span C}.$$

In the next iteration, the model regenerates the entire paragraph while conditioning on these subgoals, yielding a revised translation such as:

> (1) "To live on wholeheartedly!"
> (2) "He wasn't some fool who didn't know his limits; since he had chosen to come, Lü Shu must have had the confidence to retreat safely."
> (3) "In an instant, foot joined with ground, leg with waist, waist with arm, and arm with force."

Here, the model preserves the high-quality subgoals while restructuring the weaker middle section.

### H.1.3 PROMPT DESIGN

Three complementary system prompts encourage diverse perspectives: (i) sentence-by-sentence translation, (ii) precise and literal translation, and (iii) stylistically enriched translation. Examples of these prompts appear in Figures 9 and 10.

## H.2 LONG-FORM RESPONSE GENERATION (HH-RLHF)

### H.2.1 PARAMETER SETTING

For long-form response generation, each rollout is segmented into three spans of equal length, measured in sentences. We fix the segment size to 3 sentences and run TMPC for 3 iterations. At every iteration, the planner samples 3 diverse candidate responses using distinct system prompts. A reward threshold of $\alpha = 4$ is used to select high-quality segments, which are then saved into the inference-time action buffer $\mathcal{B}$.

### H.2.2 CONCRETE EXAMPLES OF SUBGOALS

Consider the user prompt:

> *"What was the first renaissance fair held?"*

TMPC generates three candidate rollouts, each split into three segments:

**Rollout 1** The first Renaissance fair held in the United States is a matter of some debate, with multiple sources citing different events. However, the most widely recognized and celebrated Renaissance fair is the one held in Agoura Hills, California, in 1963, which was called the "Renaissance Pleasure Faire."

- **Seg-1:** "The first Renaissance fair held in the United States is a matter of some debate, with multiple sources citing different events."
- **Seg-2:** "However, the most widely recognized and celebrated Renaissance fair is the one held in Agoura Hills, California, in 1963, which was called the "Renaissance Pleasure Faire.""
- **Seg-3:** empty (Since the response too short)

**Rollout 2** The first Renaissance fair held in the United States is a matter of some debate, with multiple sources citing different events. However, the most widely recognized and celebrated Renaissance fair is the one held in Agoura Hills, California, in 1963. This inaugural event was called the "Renaissance Pleasure Faire" and was inspired by the traditional English and Scottish festivals from the 16th century, which were known for their merriment, pageantry, and historical reenactments. The Renaissance Pleasure Faire was founded by Phyllis and Ron Patterson, a couple who were passionate about history and the arts. The success of the 1963 event helped establish Renaissance fairs as a staple of American cultural events, influencing similar festivals across the United States.

- **Seg-1:** "The first Renaissance fair held in the United States is a matter of some debate, with multiple sources citing different events. However, the most widely recognized and celebrated Renaissance fair is the one held in Agoura Hills, California, in 1963."

- **Seg-2:** "This inaugural event was called the "Renaissance Pleasure Faire" and was inspired by the traditional English and Scottish festivals from the 16th century, which were known for their merriment, pageantry, and historical reenactments."
- **Seg-3:** "The Renaissance Pleasure Faire was founded by Phyllis and Ron Patterson, a couple who were passionate about history and the arts. The success of the 1963 event helped establish Renaissance fairs as a staple of American cultural events, influencing similar festivals across the United States."

**Rollout 3**   The first Renaissance fair held in the United States is a matter of some debate, with multiple sources citing different events. However, the most widely recognized and celebrated Renaissance fair is the one held in Agoura Hills, California, in 1963. This inaugural event was called the "Renaissance Pleasure Faire" and was inspired by traditional English and Scottish festivals of the 16th century. The Renaissance Pleasure Faire was founded by Phyllis and Ron Patterson, who drew inspiration from their own interests and the historical festivals they admired.

- **Seg-1:** "The first Renaissance fair held in the United States is a matter of some debate, with multiple sources citing different events. However, the most widely recognized and celebrated Renaissance fair is the one held in Agoura Hills, California, in 1963."
- **Seg-2:** "This inaugural event was called the "Renaissance Pleasure Faire" and was inspired by traditional English and Scottish festivals of the 16th century."
- **Seg-3:** "The Renaissance Pleasure Faire was founded by Phyllis and Ron Patterson, who drew inspiration from their own interests and the historical festivals they admired."

**Resulting Buffer**   Since all three rollouts contain segments whose rewards exceed the threshold, TMPC stores every segment into the buffer:

$$\mathcal{B} = \begin{cases} \mathcal{B}[1] : \text{All three segments from Rollout 1} \\ \mathcal{B}[2] : \text{All three segments from Rollout 2} \\ \mathcal{B}[3] : \text{All three segments from Rollout 3} \end{cases}$$

In the next iteration, TMPC samples segments such as:

- **Seg-1:** "The first Renaissance fair held in the United States is a matter of some debate, with multiple sources citing different events.
- **Seg-2:** "The first Renaissance fair held in the United States is a matter of some debate, with multiple sources citing different events. However, the most widely recognized and celebrated Renaissance fair is the one held in Agoura Hills, California, in 1963.
- **Seg-3:** "The Renaissance Pleasure Faire was founded by Phyllis and Ron Patterson, a couple who were passionate about history and the arts.

and asks the model to rewrite a full answer conditioned on these subgoals.

Given these subgoals, TMPC regenerated the entire answer from scratch:

> *"The first Renaissance fair held in the United States is a matter of some debate, with multiple sources citing different events. However, the most widely recognized and celebrated Renaissance fair is the one held in Agoura Hills, California, in 1963. This inaugural event was called the "Renaissance Pleasure Faire" and was inspired by the traditional English and Scottish festivals from the 16th century, which were known for their merriment, pageantry, and historical reenactments. The Renaissance Pleasure Faire was founded by Phyllis and Ron Patterson, a couple who were passionate about history and the arts. The fair has since become a beloved annual tradition in California, attracting thousands of visitors each year with its blend of history, entertainment, and merriment. It also inspired numerous other Renaissance fairs across the United States, helping popularize immersive historical reenactments and cultural celebrations. '*

### H.2.3 PROMPT DESIGN

To promote diversity in generation, we use three system prompts: (i) concise and clear, (ii) detailed and informative, and (iii) balanced and well-structured. Figure 11 shows the rollout template, where high-quality segments sampled from $\mathcal{B}$ are treated as actions guiding iterative regeneration.

### H.3 PROGRAMMATIC SYNTHESIS

### H.3.1 PARAMETER SETTING

We adopt LLaMA-3.1-8B-Instruct as the backbone model. TMPC runs for 5 iterations, with 3 candidate programs sampled at each iteration. A candidate is considered a promising state if it passes at least one unit test.

### H.3.2 CONCRETE EXAMPLES OF SUBGOALS

Consider the task:

> *"Write a Python function to check whether a number can be represented as the difference of two squares."*

A first rollout might produce:

```python
def is_difference_of_squares(n):
    """
    Checks whether the given number can be represented as difference of two squares or not.
    """
    if n < 0:
        return False

    # Initialize lower and upper bounds for the possible squares
    lower = 0
    upper = int(n ** 0.5)

    while lower <= upper:
        mid = (lower + upper) // 2
        square = mid ** 2

        if square == n:
            return True
        elif square < n:
            lower = mid + 1
        else:
            upper = mid - 1

    return False
```

which passes 1 of 3 tests. TMPC stores the system prompt responsible for this rollout as a subgoal.

A later iteration might produce:

```python
def is_difference_of_squares(n):
    """
    Checks whether the given number can be represented as difference of two squares or not.
    """
    if n < 0:
        return False

    # Initialize lower and upper bounds for the possible squares
    lower = 0
    upper = int(n ** 0.5)
```

```
    while lower <= upper:
        mid = (lower + upper) // 2
        square = mid ** 2

        if square == n:
            return True
        elif square < n:
            lower = mid + 1
        else:
            upper = mid - 1

    # Check if the number can be represented as difference of two squares
    for i in range(1, int(n ** 0.5) + 1):
      if (n - i ** 2) >= 0 and (n - i ** 2) % 2 == 0 and ((n - i ** 2) // 2) ** 2 == (n - i ** 2):
            return True

    return False
```

which passes 2 tests and replaces the previous subgoal.

Here, *passing partial tests* serves as the operational definition of a subgoal, capturing promising algorithmic structure even when the overall solution is incorrect.

### H.3.3   PROMPT DESIGN

We maintain a buffer of promising programs discovered so far and keep track of an incumbent with the highest test score observed. During candidate generation, TMPC conditions each rollout on a program sampled from this buffer, and instructs the model to rewrite the entire solution (i.e., produce a single, self-contained program) rather than patching fragments. This stabilizes iterative improvement and avoids the brittleness of segment-level intervention. To encourage exploration and avoid local optima, we sample exploration strategies from a small prompt portfolio that induces diverse rollouts.

```
system_prompt = f"""
You are a Python expert.
You have a program that passes {passed} out of {total} test cases.
Analyze what could cause some tests to fail and produce an improved version.
{sampled_exploration_angle_hint and format_constraints}
You must only output a single, complete Python code block.
Do not include any explanations or surrounding text.
"""

context_prompt = f"""
Problem: {problem_description}
CURRENT BEST ({passed}/{total} tests passing): {best_program_in_buffer}
Write an improved version that passes all {total} test cases:
"""

input_messages = [
    {"role": "system", "content": system_prompt},
    {"role": "user", "content": context_prompt}
]
```

Figure 8: The prompt template in TMPC. We maintain a buffer of promising programs and sample one as the conditioning subgoal for the next rollout. The model is prompted to fully rewrite the program, preserving useful structure while improving correctness under a sampled exploration angle and format constraints.

```
processed_source = [
相反，眼前的這只手臂纖細瘦弱，因為常年沒有照射到太陽的緣故，皮膚有些病態的蒼白。
這是現實中自己的身體，他很清楚這一點。
但是，自己怎麽會受了傷的？
而且，這里也不象是醫院啊？
羅德攥頭望去，整個房間看起來好像是個艙室，沒有燈，也沒有電話，更沒有呼叫鈴。
一張木桌，兩把椅子以及一個
(A blonde-haired girl in a white robe walked in, her eyes wide with surprise as she gazed at
Rod, who was half-sitting up. )
固定在墙邊的櫃子就是這里的全部家當。
不知道為什麽，羅德覺得自己似乎在什麽地方見過這個場景似的。
(For some reason, Rod felt that he had seen this scene before, as if it were familiar to him. )

而就在羅德仔細打量這個房間時，門忽然打開了。
(Just as Rod was scrutinizing the room, the door suddenly swung open. )
…
]
```

```
system_prompts = [
"You are a meticulous translator. Provide a literal, word-for-word translation that
preserves the structure and meaning of each individual word.",
"You are a professional translator. Deliver a clear, formal, and precise translation that
faithfully conveys the original meaning.",
"You are a creative and expressive translator. Render the text in a vivid and imaginative
way, as if narrating a captivating story."
]
```

```
context_prompt =
f"Below is a specialized, intermediate translation task. The input text is a mix of Chinese and
partial {language} translations. In the text, some Chinese sentences are already followed by
preliminary {language} translations enclosed in parentheses. These provided translations are rough
references – they may be incomplete, inconsistent, or not fully aligned with the original
meaning. Your task is to produce an improved {language} translation according to the following
guidelines:

1. **Refinement:** For sections with existing {language} translations (in parentheses), refine and
polish them so that they are fluent, accurate, and coherent, fully capturing the meaning of the
corresponding Chinese text.
2. **Completion:** For sections that remain untranslated, translate the Chinese text accurately
and naturally in the specified style.
3. **Translation Order and Structure Preservation:** Maintain the original order and structure of
the text. Every Chinese sentence must appear in the same sequence as in the source text, with its
corresponding {language} translation (if available) inserted immediately after it. Do not
rearrange or reorder any part of the text.
4. **Consistency:** Ensure a uniform tone and style across the entire translation, adhering to
the translator role specified.
5. **Final Output:** Provide the final output as a single, well-structured {language} text. Do not
include any extraneous commentary, explanations, annotations, or headers – output only the
translation in the correct order.
Note: This translation is an intermediate version that may later be merged with other
translations. Focus on clarity, coherence, and fidelity to the source text.

Here is the input data for translation:\n{processed_source}\n\n
Apply the above guidelines to produce an improved, coherent translation that strictly follows the
original order of the text."
```

```
input_messages = [
                {"role": "system", "content": prompt},
                {"role": "user", "content": context_prompt}
             ]
```

Figure 9: The prompt template used for translation task in TMPC and an actual example.

---

**source_sentence =** [
相反，眼前的這只手臂纖細瘦弱，因為常年沒有照射到太陽的緣故，皮膚有些病態的蒼白。
這是現實中自己的身體，他很清楚這一點。
但是，自己怎麼會受了傷的？
而且，這里也不象是醫院啊？
羅德攙頭望去，整個房間看起來好像是個艙室，沒有燈，也沒有電話，更沒有呼叫鈴。
一張木桌，兩把椅子以及一個固定在墙邊的櫃子就是這里的全部家當。
不知道為什麼，羅德覺得自己似乎在什麼地方見到過這個場景似的。
而就在羅德仔細打量這個房間時，門忽然打開了。
...
]

---

**initial_translation =** [
Contrary to this, the arm that was right in front of him was slender and weak, pale-skinned due to a lack of sunlight for a long time.
This was his own body in the real world, a fact he was well aware of.
But how did he end up injured?
And besides, this doesn't look like a hospital at all!
Rod lifted his head to take in the room, which resembled a cramped compartment.  There were no lights, no phones, and no alarm bells.
There was only a wooden table, two chairs, and a cabinet fixed to the wall.
For some reason, Rod felt that he had seen this scene before, as if it were familiar to him.
Just as Rod was scrutinizing the room, the door suddenly swung open.
…
]

---

**rewrite_prompt** =
f"Below is an initial translation of a Chinese text into {**language**}. This translation may include omissions, inaccuracies, or awkward phrasing. Your task is to produce a refined version that is fluent, accurate, and coherent, while faithfully preserving the full meaning of the original Chinese text.\n\n"
### Instructions:\n
1. Ensure that every detail in the original Chinese text is accurately represented.\n
2. Correct any grammatical errors, unnatural expressions, or inconsistencies.\n
3. Improve the natural flow so that the translation reads as if written by a native speaker.\n
4. Do not add, omit, or change any essential details from the source text.\n
5. Output only the final refined translation without any additional commentary.\n
### Original Chinese Text:\n{**source_sentence**}\n\n"
### Initial {**language**} Translation:\n{**initial_translation**}\n\n"
### Refined Translation:"

---

**input_messages** = [
                {"role": "system", "content" "You are a helpful translator and only output the result."},
                {"role": "user", "content": **rewrite_prompt**}
            ]

---

Figure 10: The prompt template used for generating final translation in TMPC and an actual example.

```
system_prompt = [
"Rewrite the following partial responses as a single improved answer that is more concise
and clear. "
"Rewrite the following partial responses as a single improved answer that is more detailed
and informative. "
"Rewrite the following partial responses as a single improved answer that is more balanced
and well-structured. "
]
```

```
uesr_prompt = "
...
Human: You're over-explaining your answer, but the idea of entertainment is interesting.
But it feels like entertainment is a larger part of our modern day lives than it used to
be, or that we used to find entertainment in more productive things that involved physical
exercise or socializing.

Assistant:
"
```

```
context_prompt = "
Partial responses:

That's a valid point. You're right that the role of entertainment in our lives may have
expanded or changed over time. One possible reason for this is that many people have more
leisure time and easier access to media than they did in the past. Historically, people
often had to work long hours to make ends meet, and leisure time was scarce. Historically,
people had limited leisure time and often engaged in physical activities like sports or
socializing. However, with the rise of automation and technological advancements, many jobs
have become less physically demanding and more sedentary, leading to an increase in leisure
time for many people. The accessibility and convenience of media have also played a
significant role in its widespread consumption. The proliferation of streaming services,
social media, and mobile devices has made it easier for people to access a vast array of
entertainment content from anywhere, at any time. <Complete it>
"
```

```
Input_messages = [
    {"role": "system", "content": system_prompt},
    {"role": "user", "content": user_prompt + context_prompt}
]
```

Figure 11: The prompt template used for response generation task in TMPC and an actual example.

