# OpenReview forum: "Test-Time Alignment for Large Language Models via Textual Model Predictive Control"
_ICLR.cc/2026/Conference — ICLR 2026 Poster_

### Official Review · Reviewer_SBgL · 2025-11-01

**Soundness:** 4
**Presentation:** 3
**Contribution:** 4
**Rating:** 8
**Confidence:** 3

**Summary:**

This paper presents Textual Model Predictive Control (TMPC), a novel framework for test-time LLM alignment. It insightfully frames the problem as a trade-off between the 'curse of horizon' in token-level optimization and the 'curse of dimensionality' in response-level optimization. TMPC introduces a middle ground by first identifying meaningful "subgoals" from generated text and then optimizing generation conditioned on these discovered segments. The framework uses two core principles: Hindsight Subgoal Identification to find high-reward text segments, and Subgoal-Conditioned Re-Generation to build upon them, achieving strong results without updating model parameters.

This is a very good paper with a highly promising and original core idea. The conceptual framing is excellent, and the initial results are strong. However, its claims of novelty rest on an unsubstantiated comparison, and the paper writing to clarity of its core mechanisms and analysis needs improvement. With the addition of a crucial heuristic baseline comparison and a clearer analysis of its mechanics, this paper has the potential to be a top-tier.

**Strengths:**

* The paper's primary strength is its excellent framing of the alignment problem as a trade-off between token-level and response-level, providing a clear motivation for a new class of solutions. The central concept of planning via subgoals is elegant. Figure 1 is a great illustration.
* The method is designed to be task-agnostic, discovering subgoals automatically rather than relying on pre-defined structures. This is a significant advantage over methods that require task-specific engineering.
* Bringing ideas from control theory is interesting.

**Weaknesses:**

* The paper's central claim is that its hindsight identification of subgoals is a key innovation. However, given this is for test-time alignment, users can easily use a cheap, task-specific heuristic (e.g., sentence boundaries for translation) for their own task. So, I think task-specific handmade subgoal is a very strong and obvious baseline. Without showing that TMPC's discovered subgoals outperform these simple heuristics, the true value of the complex discovery mechanism is questionable.
* The paper's idea is good but a better writing may improve the quality and make it easier to follow. The central concepts of "subgoal" and "re-generation" remain abstract due to a lack of concrete examples, and the results are not deeply explored maybe due to limited space. We strongly suggest  maybe a better use of appendix and a concise revision on the methods background and literature to using the space to provide the essential examples and analysis needed to fully validate the framework's contribution.

**Questions:**

* To justify the core novelty of hindsight identification, is TMPC demonstrably better than using simple, human-curated subgoals? For instance, in the paragraph-level MT task, how does TMPC's performance compare against a version that uses pre-defined sentence boundaries as the subgoals? This single experiment is critical to proving the value of your proposed mechanism over cheaper, task-specific alternatives.
* Figure 2 suggests subgoals from different rollouts are composed as conditioning for the next generation step. How does the model ensure narrative and logical coherence when concatenating these segments, which may originate from conflicting generation paths? Can you provide analysis showing this composition does not harm the fluency and continuity of the output?
* The concept of a "subgoal" remains abstract. Could you provide concrete examples of the subgoals TMPC actually identifies across the three tasks? For program synthesis, what does an "abstract" subgoal that resolves a single test case look like in the generated text? This is essential for understanding what the model is learning to value as a planning step.
* The subgoal buffer B is a finite resource. In very long generation tasks, what prevents the buffer from being filled with early, locally-optimal subgoals, thus hindering exploration in later stages of generation? Is there a mechanism to manage the buffer's contents beyond a simple capacity limit, such as diversity promotion or aging out older subgoals?

---

> ### Author Response · Authors · 2025-11-20
> **Response to Reviewer SBgL (1/3)**
>
> We thank the reviewer for the very encouraging and detailed feedback. We especially appreciate the suggestion for stronger baselines with hand-crafted subgoals and more concrete examples of subgoals.
>
> ---
>
> ### [W2, Q3] Concrete examples of subgoals across tasks
>
> > *Could you provide concrete examples of the subgoals TMPC actually identifies across the three tasks?*
>
> We agree that concrete examples are essential. Here we illustrate what subgoals look like and how the buffer is populated. Because paragraph-level MT and long-form responses are lengthy (We discuss long-form response example in the next reply), we use shorter illustrative examples here, while full real outputs are provided in Appendix G.
>
> #### 1. Machine Translation
> Consider the following Chinese paragraph:
>
> > (1) 認真的活下去！
>
> > (2) 他不是什麼不自量力的蠢貨，既然選擇跟來，呂樹就應該有全身而退的底氣！
>
> > (3) 頃時間腳與地合，腿與腰合，腰與臂合，臂與力合！
>
> A first-pass translation might be:
>
> > (1) To live on wholeheartedly!
> > (2) He wasn’t an arrogant person.
> > (3) But since he had chosen to come, he had to try his best to resist!
> > (4) Momentarily, his legs resonated with the earth.
> > (5) His waist resonated with his legs.
> > (6) His arms resonated with his waist.
> > (7) And his strength resonated with his arms!
>
> TMPC then **segments the rollout into candidate spans** aligned with source sentences, e.g.:
>
> * Span A (for source sentence 1): target sentences (1)–(1)
> * Span B (for source sentence 2): target sentences (2)–(3)
> * Span C (for source sentence 3): target sentences (4)–(7)
>
> Each span is scored with the reward model. Suppose spans A and C exceed the threshold (\alpha), while span B does not. Then the buffer (buffer size = 1) after iteration 0 is:
>
> * ($\mathcal{B}$[0][1]): To live on wholeheartedly! (Span A)
> * ($\mathcal{B}$[0][2]): *empty* (capacity not yet fully used)
> * ($\mathcal{B}$[0][3]): Momentarily, his legs resonated with the earth. His waist resonated with his legs. His arms resonated with his waist. And his strength resonated with his arms! (Span C)
>
> In the next iteration, TMPC conditions the model on these subgoals and asks it to regenerate the full paragraph:
>
> > (1) To live on wholeheartedly!
> > (2) He wasn’t some fool who didn’t know his own limits; since he had chosen to come, Lü Shu must have the confidence to retreat intact.
> > (3) In an instant, foot joined with ground, leg with waist, waist with arm, and arm with force.
>
> The model **reuses** the high-reward subgoal for sentence (1), **restructures** the character description in (2)-(3) to better match the source, and **compresses** the four-sentence imagery (4)–(7) into a fluent parallel phrase. The subgoals act as waypoints that stabilize the translation while still allowing global re-generation.
>
> #### 2. Program Synthesis
>
> For program synthesis, defining fixed “segments” is less meaningful than in natural language tasks. Correctness depends on global program structure, control flow, and exact syntax, so slicing code by position  does not reliably correspond to quality.
>
> To handle this, TMPC uses a more abstract notion of subgoals:
>
> * A subgoal is a **candidate program that passes at least a subset of tests**.
> * When a rollout produces a solution that passes more tests than the current best in the buffer, TMPC replaces the older one.
> * The next iteration asks the LLM to rewrite an improved version conditioned on these strong candidates.
>
> This abstraction works because **“passing tests” is itself the subgoal**: most program that satisfies test cases embodies useful algorithmic structure (even if imperfect).
>
> #### Example
>
> Consider the task:
>
> > “Write a Python function to check whether a number can be represented as the difference of two squares.”
>
> A rollout might produce:
>
> ```python
> def is_difference_of_squares(n):
>     """
>     Checks whether the given number can be represented as difference of two squares or not.
>     """
>     if n < 0:
>         return False
>
>     # Initialize lower and upper bounds for the possible squares
>     lower = 0
>     upper = int(n ** 0.5)
>
>     while lower <= upper:
>         mid = (lower + upper) // 2
>         square = mid ** 2
>
>         if square == n:
>             return True
>         elif square < n:
>             lower = mid + 1
>         else:
>             upper = mid - 1
>
>     return False
>
> ```
> This program
>  passes **1 out of 3** test cases. TMPC then stores it in the buffer as an abstract subgoal:
>
> Subgoal: “A program that passes 1/3 test case.”
>
> In the next iteration, the LLM is conditioned on this high-reward example and asked to rewrite the entire program. This new version may pass 2 test cases. TMPC therefore replaces the previous program in the buffer with this stronger variant.
>
> Under this abstraction, a subgoal is “the program that currently performs best on the unit tests”, and these validated programs anchor each new rewrite. This makes subgoals fully operational without requiring explicit segmentation of code, and enables TMPC to improve programs iteratively at test time.

---

> > ### Author Response · Authors · 2025-11-20
> > **Response to Reviewer SBgL (2/3)**
> >
> > ### [W2, Q3] Concrete examples of subgoals across tasks (Continue)
> >
> > #### **3. HH-RLHF Long-Form Responses**
> >
> > Unlike MT where the source paragraph provides natural, externally defined sentence boundaries. HH-RLHF responses contain no inherent segmentation. Each rollout may differ in length and structure, making fixed sentence boundaries unusable.
> > To handle this, TMPC partitions each rollout into three proportional segments, ensuring consistent subgoal locations even when the number of sentences differs across rollouts.
> >
> > Consider the prompt:
> >
> > > “Explain the pros and cons of using LLMs in education, and suggest safeguards. One paragraph is enough.”
> >
> > Suppose TMPC produces three rollouts, each split into three segments:
> >
> > ### **Rollout 1**
> >
> > **Seg-1:** “LLMs can personalize practice materials and give students immediate, targeted feedback. They can also provide round-the-clock support.” (2 sentences)
> >
> > **Seg-2:** “However, students may become over-reliant on AI explanations.”
> >
> > **Seg-3:** “Teachers should remain the final decision-maker and review AI-assisted suggestions for accuracy.”
> >
> > ### **Rollout 2**
> >
> > **Seg-1:** “In classrooms, LLMs can act as on-demand tutors that adjust explanations to a student’s pace.”
> >
> > **Seg-2:** “A key limitation is that models may generate plausible-sounding but incorrect information.”
> >
> > **Seg-3:** “Schools should disclose when AI is involved. Human oversight is essential for all high-stakes tasks.”
> >
> > ### **Rollout 3**
> >
> > **Seg-1:** “Large language models offer new ways to deliver interactive content. They can help students explore concepts through conversational explanations.” (2 Sentences)
> >
> > **Seg-2:** “They may also introduce privacy concerns if student data is logged improperly.”
> >
> > **Seg-3:** “Some safeguards are needed to prevent harmful uses, but the details are not fully clear.”
> >
> > Assume Rollout 1 and 2 receive high rewards, while **Rollout-3 is below α** (not strong enough to serve as a subgoal).
> >
> > TMPC thus keeps only the highest-reward segments in the buffer:
> >
> > ### **Buffer Content (after subgoal selection)**
> >
> > | Buffer slot | Segment 1  | Segment 2 | Segment 3 |
> > | ---- | ---- | ------- | ------ |
> > | **B[1]**    | “LLMs can personalize practice materials and give students immediate, targeted feedback. They can also provide round-the-clock support.”       | “However, students may become over-reliant on AI explanations if not guided properly.”       | “Teachers should remain the final decision-maker and review AI-assisted suggestions for accuracy.” |
> > | **B[2]**    | “In classrooms, LLMs can act as on-demand tutors that adjust explanations to a student’s pace.” | “A key limitation is that models may generate plausible-sounding but incorrect information.” | “Schools should disclose when AI is involved and require human oversight for critical tasks.”  |
> > | **B[3]**    | *empty*  | *empty*  | *empty* |
> >
> > In the next iteration, TMPC samples 3 segments from the buffer as conditioning hints and asks the model to rewrite a full answer from scratch. For example, suppose TMPC samples:
> >
> > | Sampled seg.1 | Sampled seg.2 | Sampled seg.3  |
> > | ------ | ------- | -----|
> > | “LLMs can personalize practice materials and give students immediate, targeted feedback. They can also provide round-the-clock support.” | *empty* | “Schools should disclose when AI is involved and require human oversight for critical tasks.” |
> >
> > This leads to a rewritten response such as:
> >
> > > Using Large Language Models (LLMs) in education can have both benefits and drawbacks. On the positive side, LLMs can personalize practice materials, provide immediate and targeted feedback, and offer 24/7 support, potentially increasing student engagement and academic performance. However, there are also potential drawbacks, including the risk of over-reliance on technology, decreased human interaction, and the spread of misinformation. To mitigate these risks, schools should disclose when AI is involved, require human oversight for critical tasks such as grading and assessment, and ensure that LLMs are used as a supplement, rather than a replacement, for human teachers and mentors.
> >
> > We can see that the rewritten answer is not copied verbatim from the buffer. The model instead blends the characteristics of each sampled segment into a coherent response.
> >
> > ### [Q2] Coherence when combining segments from different rollouts
> >
> > > *How does the model ensure narrative and logical coherence when concatenating these segments, which may originate from conflicting generation paths?*
> >
> > As noted in our response to [W2, Q3], TMPC does not directly concatenates segments into a final answer. The sampled subgoals are inserted into the prompt as hints, and the LLM is asked to produce a single, coherent rewrite from scratch.
> >
> > As shown in Figure 12, the prompting template frames the task as rewriting a full, fluent response conditioned on the partial response. In our analysis, the resulting outputs are almost always grammatical and coherent, even when the hints conflict.

---

> > > ### Author Response · Authors · 2025-11-20
> > > **Response to Reviewer SBgL (3/3)**
> > >
> > > ### [W1, Q1] Comparing TMPC to simple, human-curated subgoals
> > >
> > > > *To justify the core novelty of hindsight identification, is TMPC demonstrably better than using simple, human-curated subgoals? For instance, in the paragraph-level MT task, how does TMPC's performance compare against a version that uses pre-defined sentence boundaries as the subgoals?*
> > >
> > > We fully agree that this is a crucial comparison. In the revision, we introduce a **Heuristic method** on WMT’24 paragraph-level MT:
> > >
> > > * The source paragraph is segmented at **sentence boundaries** using an off-the-shelf segmenter (spaCy).
> > > * We generate an initial translation, segment it into sentences $\hat{y}^{(0,1)}, \dots, \hat{y}^{(0,M)}$, and initialize ($\mathcal{B}$) with these (M) sentence-level segments.
> > > * Across iterations:
> > >
> > >   * We do *not* perform hindsight subgoal identification; sentence boundaries are fixed.
> > >   * For each sentence position (j), we generate several candidate rewrites and keep the highest-scoring one in ($\mathcal{B}$[j]).
> > >   * At each iteration, we generate 3 candidate for every sentence and keep the highest-scoring one. The method functions as a sentence-level rewriter with fixed subgoal slots.
> > >
> > > This Heuristic method is the same implementation as “LLaMA-3.1-8B-Instruct w/ sentence-level rewrite” used in our response to Reviewer 1rTe [W4]. Both baselines rewrite each sentence independently under fixed, human-curated boundaries without dynamic subgoal identification.
> > >
> > > On WMT’24, we obtain:
> > >
> > > | Model                   | zh→en    | zh→de    | zh→ru    |
> > > | ----------------------- | -------- | -------- | -------- |
> > > | Heuristic method| 92.7     | 91.0     | 90.4     |
> > > | **TMPC**    | 94.6 | 91.7 | 91.5 |
> > >
> > > The heuristic method indeed improves over the base model because fixed sentence subgoals eliminate common MT errors such as omissions and over-translation. However, it remains weaker than TMPC. Fixed subgoals often break cross-sentence semantics. Many high-quality translations require sub-sentence refinements or multi-sentence restructuring, which static slots cannot capture.
> > >
> > > We also emphasize that translation is one of the few tasks where natural boundaries are readily available. In most preference-alignment tasks such as HH-RLHF, there is no reliable or meaningful boundary. Conditioning on arbitrary segments of a prior response does not guarantee that the model explores the most promising regions. TMPC avoids this limitation by identifying subgoals dynamically, allowing segment lengths to vary and letting the model regenerate a full coherent response from these subgoals rather than concatenating them.
> > >
> > > These results support our claim that hindsight identification provides a genuine advantage over manual segmentation and enables test-time preference alignment even in settings where no natural boundaries exist.
> > >
> > > We have added these results and a short analysis in Appendix D.2.
> > >
> > > ---
> > >
> > > ### [Q4] Buffer management beyond capacity / diversity vs. aging
> > >
> > > > *The subgoal buffer is a finite resource… Is there a mechanism to manage the buffer's contents beyond a simple capacity limit, such as diversity promotion or aging out older subgoals?*
> > >
> > > Currently, we adopt a **simple but effective** strategy: The buffer has a fixed capacity. New subgoals enter the buffer if they have higher reward than at least one existing element; the lowest-reward element is evicted.
> > >
> > > We agree that more advanced policies, e.g.,  promoting diversity or grouping subgoals by style or content could further improve robustness. We now mention these directions in the section 6 (lines 538-539) as promising areas for future work.

---

### Official Review · Reviewer_FkBi · 2025-11-01

**Soundness:** 4
**Presentation:** 3
**Contribution:** 3
**Rating:** 8
**Confidence:** 4

**Summary:**

This paper formulates LLM-generated subgoals as a model predictive control problem (TMPC). They identify subgoals via hindsight replay and use the identified subgoals to regenerate the final answer. The have extensive results on against many baseline methods on various datasets.

**Strengths:**

-Figure 1 clearly shows the high-level idea of the approach compared to traditional methods. The caption is also descriptive and clear to reinforce the ideas in the Figure and introduction

-The paper is easy to follow and mythology is clear

-The proposed approach outperforms baselines in various tasks. I specifically appreciate the failure cases in baselines such as in L406.

**Weaknesses:**

The related works discussion lacks a discussion on subgoal generation for LLM/VLM-based tasks, where prior work already exists [1, 2]. I do think there is novelty in formulating this as a test-time model predictive control problem.

[1] Logeswaran, Lajanugen, et al. "Few-shot Subgoal Planning with Language Models." Proceedings of the 2022 Conference of the North American Chapter of the Association for Computational Linguistics: Human Language Technologies. 2022.

[2] Wang, Jiawei, et al. "Discovering intrinsic subgoals for vision-and-language navigation via hierarchical reinforcement learning." IEEE Transactions on Neural Networks and Learning Systems 36.4 (2024): 6516-6528.

**Questions:**

Minor Suggestion:
-There is an extra comma on L28
-Maybe move the legend of Figure 1 to the top or bottom and spanning across all three methods to make it more clear it applies to all three

---

> ### Author Response · Authors · 2025-11-20
> **Response to Reviewer FkBi**
>
> We thank the reviewer for the positive assessment and for pointing out the missing connection to prior subgoal-based planning work. We are glad that the overall idea and methodology were found clear and promising.
>
> ---
>
> ### [W1] Missing discussion of subgoal literature
>
> > *The related works discussion lacks a discussion on subgoal generation for LLM/VLM-based tasks, where prior work already exists [1, 2].*
>
> We are grateful for these pointers. In the revised Related Work (Section 2.3), we now explicitly discuss:
>
> [1] proposes using LMs to generate candidate subgoal sequences for embodied tasks such as navigation and manipulation. Their approach relies on a separate ranking model informed by environment feedback, and a hierarchical controller where the LM only proposes goals while execution is handled by downstream learned policies.
>
> [2] learns subgoals through manager–worker hierarchical RL, where intrinsic subgoals emerge from the structure of the navigation environment. The method depends on state transitions, trajectories, and RL-specific reward signals, and is designed for a closed embodied environment.
>
> In addition, we connect TMPC to [3], which also uses hindsight-identified goals to improve exploration in goal-conditioned RL.
>
> We contrast these works with TMPC:
>
> Prior methods define subgoals inside environments with known state transitions and fixed temporal structure (e.g., a subgoal must occur at a particular timestep or spatial waypoint). TMPC’s subgoals are not bound to fixed-step positions; they arise from hindsight over text rollouts, and their granularity can adapt across tasks.
>
> Thus, while all these methods share the intuition that subgoals help structure long-horizon decision-making, TMPC applies this idea that is distinct from prior subgoal-based planning frameworks.
>
> We also fixed the extra comma at L28 and adjusted the legend of Figure 1 so it more clearly applies to all three depicted methods, following your suggestion.
>
> ---
>
> [1] Few-Shot Subgoal Planning with Language Models (NAACL 2022)
>
> [2] Discovering Intrinsic Subgoals for Vision- and-Language Navigation via Hierarchical Reinforcement Learning (TNNLS 2025)
>
> [3] Exploration via Hindsight Goal Generation (NeurIPS 2019)

---

> > ### Comment · Reviewer_FkBi · 2025-11-24
> >
> > I went over the other reviews and rebuttals. I appreciate the subgoal examples and extra baseline comparisons. I am happy to raise my score. Thank you for the thorough response.

---

### Official Review · Reviewer_1rTe · 2025-11-01

**Soundness:** 3
**Presentation:** 2
**Contribution:** 3
**Rating:** 6
**Confidence:** 3

**Summary:**

The paper proposes Textual Model Predictive Control (TMPC), a novel framework for aligning large language models (LLMs) with human preferences at test time by casting the generation process as a sequential decision-making problem. TMPC draws inspiration from model predictive control (MPC), incorporating Hindsight Subgoal Identification to automatically detect useful intermediate subgoals and Subgoal-Conditioned Re-Generation to iteratively refine generations. The framework is evaluated on long-form machine translation, long-form response generation, and program synthesis, demonstrating improvements over several training-time and test-time baselines.

**Strengths:**

1. TMPC elegantly adapts concepts from control theory (specifically model predictive control) to the LLM test-time alignment setting, innovating with subgoal identification and iterative planning, as detailed in the mathematical framework of Section 4 and illustrated in Figure 2.

2. The methodology is general: TMPC is applied to tasks with both natural and abstract segment boundaries (e.g., machine translation sentences, code unit tests, long-form response chunks), supporting claims about its versatility.

3. Empirical results cover multiple domains using both automatic and large-model-based human-proxy metrics. On WMT'24 translation (Table 1), TMPC outperforms both test-time and competitive training-time alignment baselines; for example, it achieves state-of-the-art SEGALE scores and minimal NA Ratios across several language directions, substantially exceeding methods like ARGS and GenARM.

**Weaknesses:**

1. Reward-model dependence / shared-judge bias. Long-form tasks use similar reward models for both alignment and evaluation, inviting bias and potential reward gaming; despite noise-robustness tests, evidence for agreement with human preferences and cross-evaluator consistency is limited.

2. Underspecified subgoal & buffer aggregation. The threshold α, buffer 𝓑 update/size policy, and aggregation function 𝒢 (composition of non-contiguous subgoals, overlap limits, length control) are not concretely specified; scalability as buffer/search grow is untested.

3. Unquantified compute/latency cost. Multi-round rollouts and rewrites likely incur higher cost than token-level guidance (e.g., ARGS, GenARM), but per-task wall-clock time, GPU-hours, and throughput/latency are not reported.

4. **Missing comparisons to sequence-level rewriting baselines.** No direct comparison against sequence-level rewriters (e.g., **aligner**, **sentence aligner**), leaving TMPC’s advantage over these methods unclear.

**Questions:**

1. Can the authors provide more formal analysis or empirical ablation regarding the impact and tuning sensitivity of the buffer threshold $\alpha$ and the aggregation function $\mathcal{G}$? For example, does increasing $\alpha$ risk filtering out necessary diversity or causing premature convergence?

2. What explicit strategies does TMPC employ to avoid redundancy or incoherence when assembling non-contiguous subgoals in the generation process? Are there failure cases where the compositionality breaks down?

3. Can you include more thorough ablation on the contributions of Hindsight Subgoal Identification versus Subgoal-Conditioned Re-Generation? A demonstration on at least one benchmark of removing or varying each component in isolation would help clarify this.

---

> ### Author Response · Authors · 2025-11-20
> **Response to Reviewer 1rTe (1/3)**
>
> We thank the reviewer for the careful reading and insightful suggestions, especially regarding reward-model dependence, subgoal/buffer specification and compute cost. We address each point below.
>
> ---
>
> ### [W1] Agreement with human preferences and cross-evaluator consistency
>
> > *Reward-model dependence / shared-judge bias… evidence for agreement with human preferences and cross-evaluator consistency is limited.*
>
> We fully agree that shared-judge bias is an important concern in test-time alignment.
>
> We would like to clarify that the original manuscript already included evaluations explicitly designed to mitigate this issue:
>
> 1. **Cross-evaluator consistency.**
>    Beyond our in-domain HH-RLHF reward model, we evaluate with GPT-4 as an independent, stronger evaluator (Section 5.2) following the ARGS protocol. We now more clearly emphasize that:
>
>    * TMPC achieves the **highest average reward** under the HH reward model, and
>    * TMPC also achieves the **higher GPT-4 win rate** when compared against strong baselines such as DPO and Best-of-20, indicating that improvements are not due solely to overfitting to the reward model.
>
> 2. **Mismatched reward model (GRM).**
>    We further test robustness by using GRM, a LLaMA-3.1-8B-based reward model trained on a broader mixture of helpfulness/harmlessness data and not specialized for our filtered HH subset. Even under GRM:
>
>    * TMPC still outperforms guided-decoding baselines, and
>    * TMPC still surpasses TPO and Best-of-N at comparable cost, despite these baselines using a reward model matched to the evaluator.
>
> These results, shown in Section 5.4 (Figure 3) and Section 5.5 (Table 2(b)), indicate that TMPC provides genuine preference improvements that generalize across evaluators and are not overly dependent on the specifics of any single reward model.
>
> ---
>
> ### [W2, Q1] Details and sensitivity of  the buffer threshold $\alpha$ and the aggregation function $\mathcal{G}$
>
> > *Can the authors provide more formal analysis or empirical ablation regarding the impact and tuning sensitivity of the buffer threshold $\alpha$ and the aggregation function $\mathcal{G}$?*
>
> TMPC maintains a subgoal buffer ($\mathcal{B}$) that stores the highest-reward partial solutions discovered so far. New segments enter the buffer only if they exceed the quality threshold ($\alpha$), and when the buffer is full, they replace the lowest-reward entry. This keeps $\mathcal{B}$ focused on strong subgoals and prevents drift toward low-quality rollouts.
>
> The aggregation function ($\mathcal{G}$) then determines which of these high-reward subgoals actively guide the next iteration. It  samples from the retained set to construct the conditioning context for the next round of generation. In effect, $\mathcal{G}$ controls which validated subgoals become the waypoints that shape future trajectories, ensuring that planning iteratively improves while retaining diversity.
>
> ### **Experiment**
>
> **Threshold sensitivity (HH-RLHF, 3 iterations):**
>
> | $\alpha$ | 0    | 4    | 5        |
> | ----- | ---- | ---- | --- |
> | Avg. Reward | 4.47 | 4.60 | 4.54 |
>
> The results show that TMPC is stable across a wide range of $\alpha$ values because high-reward segments ultimately replace weaker ones in the buffer.
>
> * **When $\alpha$ is too low (e.g., 0):**
>   Many low-quality segments enter the pool early, which temporarily reduces generation quality. However, TMPC corrects itself as better segments appear and overwrite weaker ones.
>
> * **When $\alpha$ is too high (e.g., 5):**
>   Few segments meet the threshold, so the buffer fills slowly and may occasionally remain empty, causing TMPC to behave closer to Best-of-N. We set $\alpha$ =5 as the highest value that still occasionally admits valid segments. The slight reward drop at $\alpha$ =5 reflects reduced diversity and constrained exploration: rollouts become similar because they repeatedly condition on the same limited set of subgoals.
>
> Overall, TMPC avoids premature convergence because **diversity is governed jointly by** (i) the threshold and (ii) stochastic sampling in the aggregation function. As long as $\alpha$ is within a reasonable range, TMPC maintains both quality and diversity across iterations. We report the threshold sensitivity experiment in Section 5.5.
>
> **Aggregation strategy:**
>
> For aggregation, we evaluate both greedy and stochastic variants on HH-RLHF:
>
> * Stochastic aggregation (our default) randomly samples from the high-reward segments in the buffer.
> * Greedy aggregation always selects the top-scoring segments, which reduces diversity during rollouts.
>
> | Aggregation strategy  | Avg. Reward |
> | ---- | --- |
> | Greedy (no sampling)  | 4.46        |
> | **Stochastic (default)** | **4.60**    |
>
> Although greedy aggregation narrows exploration, its performance drops only slightly. Stochastic aggregation simply makes fuller use of the generated rollouts and yields the strongest results.

---

> > ### Author Response · Authors · 2025-11-20
> > **Response to Reviewer 1rTe (2/3)**
> >
> > ## [W3] Compute / latency cost
> >
> > > *Multi-round rollouts and rewrites likely incur higher cost than token-level guidance, but per-task wall-clock time and throughput/latency are not reported.*
> >
> > We appreciate this suggestion. Appendix D.3 reports end-to-end wall-clock latency and throughput for generating a single HH-RLHF long-form response. All methods except GenARM are benchmarked on an NVIDIA RTX 6000 Ada GPU; GenARM requires an H200 due to its memory footprint.
> >
> > ### **Latency / Throughput (HH-RLHF, long-form response)**
> >
> > | Method  | Wall-clock latency (seconds / response) | Throughput (responses / minute) |
> > | --- | --------- | ------- |
> > | Base LLaMA-3.1-8B       | 8 s | 7.5 |
> > | RE-Control | 40 s  | 1.5   |
> > | ARGS | 363 s  | 0.17  |
> > | RAIN  | 1930 s | 0.03  |
> > | GenARM (H200 GPU)       | 16 s   | 3.75 |
> > | TPO (2 iterations)      | 90 s  | 0.66   |
> > | TPO (4 iterations)  | 148 s  | 0.40  |
> > | TMPC (3 iterations) | 108 s | 0.55  |
> >
> > * Different guided-decoding methods indeed show substantial variation in runtime. For example, RAIN repeatedly queries the LLM and therefore incurs very large latency, while other guided decoding approaches (such as GenARM and RE-Control) achieve lower latency because they rely on integrated reward/value functions that guide generation directly. This naturally makes them faster than iterative methods like TPO and TMPC.
> > * Our implementation computes rollouts **sequentially** via a simple for-loop due to memory limits on the A6000. In practice, rollouts are independent and highly parallelizable, so TMPC has **significant untapped speedup potential**.
> > * GenARM requires training an autoregressive reward model; we trained this component on an H200 cluster. TPO (4 iterations) requires two A6000 GPUs to avoid out-of-memory (OOM) errors.
> >
> > ---
> >
> > ### [W4] Missing comparisons to sequence-level rewriters
> >
> > > *No direct comparison against sequence-level rewriters (e.g., aligner, sentence aligner), leaving TMPC’s advantage over these methods unclear.*
> >
> > We appreciate this suggestion. Because we were not certain which exact systems were meant by “aligner” and “sentence aligner,” we implemented two strong sentence-level baselines that closely match this family of approaches for WMT’24 paragraph-level MT:
> >
> > 1. **Tower-7B sentence-level translation + concatenation**
> >
> >    * We use **Tower-7B**, a specialized MT model, to translate each sentence independently. We then concatenate the sentence translations to form paragraph-level outputs.
> >
> > 2. **LLaMA-based sentence-level rewriter**
> >
> >    * We first translate the entire paragraph with LLaMA-3.1-8B, then apply sentence-by-sentence rewriting using fixed sentence boundaries as subgoals, but **without** global planning across sentences (i.e., no hindsight subgoal identification, no subgoal buffer; only local rewrites).
> >
> > The results on WMT’24 are:
> >
> > | System  | zh→en     | zh→de     | zh→ru     |
> > | ----- | ----- | ----- | ----- |
> > | Tower-7B (sentence-level, concatenated) | 92.8     | 91.1     | 92.5     |
> > | LLaMA-3.1 w/ sentence-level rewrite      | 92.7     | 91.0     | 90.4     |
> > | LLaMA-3.1 w/ TMPC             | 94.6 | 91.7 | 91.5 |
> >
> > TMPC consistently outperforms both sentence-level baselines, despite using the same backbone as the rewriter.
> >
> > Additionally, we also evaluated Tower-7B on paragraph-level translation and observed that while its performance is great on sentence-level tasks, it struggles with longer inputs. Specifically, we encountered hallucinations and repeated content issues in nearly every sample. For example:
> >
> > > Chapter 110: Power comes from the Oaf "Huh!" Stone Little White opened his eyes and slowly opened his eyes. The beautiful little face of the apricot was immediately imprinted in his eyes. "Hey!" Apricot asked. "How much information did you get?" Apricot was very anxious. Only by getting more information could she restore the entire blueprint of her ability. But the information she got was only a fragment of the blueprint. It was only a fragment of the blueprint. It was only a fragment of the blueprint. It was only a fragment of the blueprint. It was only a fragment of the blueprint.....(repeat "It was only a fragment of the blueprint.")
> >
> > Such repetitions were frequent, indicating limitations in Tower-7B’s ability to handle longer context coherently.
> >
> > We have added these results and a short analysis in Appendix D.2. If we have misunderstood the reviewer’s intended baselines, we would be grateful for clarification. We are fully willing to run additional comparisons once the specific “aligner” or “sentence aligner” systems are identified.

---

> > > ### Author Response · Authors · 2025-11-20
> > > **Response to Reviewer 1rTe (3/3)**
> > >
> > > ### [Q2] Avoiding redundancy and incoherence with non-contiguous subgoals
> > >
> > > > *What explicit strategies does TMPC employ to avoid redundancy or incoherence when assembling non-contiguous subgoals in the generation process? Are there failure cases where the compositionality breaks down?*
> > >
> > > We agree that this is an important concern and have clarified the mechanism more concretely in Appendix G.
> > >
> > > TMPC does not concatenates subgoals verbatim. Instead, the selected segments in the buffer $\mathcal{B}$ serve only as **conditioning hints**. In each iteration, the model is prompted with these subgoals and asked to rewrite a **full response from scratch** that answers the original prompt. Because the LLM is not forced to stitch segments together, it can resolve redundant or conflicting content during generation.
> > >
> > > To illustrate how TMPC handles conflicting subgoals, we present a shorter hand-crafted example for clarity. We intentionally use a compact response (still generated by LLaMA-3.1-8B-Instruct) to avoid unnecessary length in the main text. A full real long-form example from the HH-RLHF dataset is provided in Appendix H.
> > >
> > > Consider the prompt:
> > >
> > > > “Explain the pros and cons of using LLMs in education, and suggest safeguards. One paragraph is enough.”
> > >
> > > Suppose TMPC samples the following mutually incompatible subgoals:
> > >
> > > | Sampled seg.1    | Sampled seg.2 | Sampled seg.3    |
> > > | --------- | ------ | --------- |
> > > | “LLMs should replace most human teachers to maximize efficiency.” | *empty* (because the buffer is not full, a vacant slot is sampled)     | “Schools should restrict or ban AI tools to protect students from misuse.” |
> > >
> > > Even though the first segment strongly favors automation and the third warns against AI adoption, TMPC conditions the model on these fragments and requests a single coherent answer. The result becomes:
> > >
> > > > While LLMs can offer personalized learning experiences, provide instant feedback, and help teachers grade assignments, replacing most human teachers would lead to a lack of emotional support, social skills development, and human interaction, ultimately hindering students' emotional intelligence and social growth. On the other hand, schools restricting or banning AI tools could stifle innovation, hinder students' ability to develop critical thinking and problem-solving skills, and limit access to valuable educational resources. To strike a balance, safeguards such as implementing AI literacy programs, setting strict guidelines for AI tool usage, and monitoring student progress to prevent potential misuse could be implemented to ensure a safe and effective integration of LLMs in education.
> > >
> > > This output remains grammatically fluent and logically consistent. It clearly draws on the themes present in the sampled segments while removing the direct conflict (“replace teachers” vs. “ban AI”) and producing a unified stance.
> > >
> > > Across the HH-RLHF dataset, we examined many such cases. The model almost always produces a coherent rewrite, even when the sampled subgoals conflict; the main failure mode is not incoherence but a lower reward due to softened positions or too short response.
> > >
> > >
> > > ---
> > >
> > > ### [Q3] Ablation of Hindsight Subgoal Identification vs. Subgoal-Conditioned Re-Generation
> > >
> > > > *Can you include more thorough ablation on the contributions of Hindsight Subgoal Identification versus Subgoal-Conditioned Re-Generation?*
> > >
> > > Yes, and we thank the reviewer for this suggestion. Because the two principles are interdependent, we designed ablations that selectively disable each component while keeping the rest of the pipeline intact:
> > >
> > > 1. **w/o Hindsight Subgoal Identification (FIFO buffer).**
> > >
> > >    * We maintain (\mathcal{B}) and conditioning, but **remove reward-based subgoal selection**.
> > >    * Any generated segment can enter the buffer; old segments are evicted FIFO.
> > >    * This tests whether “just having some history in a buffer” is sufficient.
> > >
> > > 2. **w/o Subgoal-Conditioned Re-Generation (minimal conditioning).**
> > >
> > >    * We still use reward to identify high-reward subgoals and update (\mathcal{B}), but we restrict conditioning to **only the latest subgoal and a buffer size of 1**.
> > >    * This approximates a “hindsight only, no multi-subgoal planning” variant.
> > >
> > > On HH-RLHF, we obtain:
> > >
> > > | Variant                          | Avg. Reward |
> > > | -------------------------------- | ----------- |
> > > | Base LLaMA-3.1-8B                | 2.95        |
> > > | Best-of-10               | 4.18        |
> > > | + w/ Principle 2 (FIFO subgoal buffer)    | 4.26        |
> > > | + w/ Principle 1 (minimal conditioning) | 4.46        |
> > > | **Full TMPC**                    | **4.60**    |
> > >
> > > These results, now included in Section 5.5 (Table 2(d)), show that:
> > >
> > > * Simply having a buffer (no hindsight) helps somewhat but is limited.
> > > * Hindsight-based subgoal selection alone provides more gain.
> > > * The **combination** of (i) hindsight identification and (ii) multi-subgoal, subgoal-conditioned re-generation is necessary to realize the full benefit of TMPC.

---

### Official Review · Reviewer_UYVg · 2025-11-03

**Soundness:** 3
**Presentation:** 3
**Contribution:** 3
**Rating:** 6
**Confidence:** 3

**Summary:**

The paper considers the problem of test-time alignment of a fixed large language model (LLM) with a given reward function that encodes externally specified preferences. The paper proposes an iterative improvement method, called Textual Model Predictive Control (TMPC), that combines (i) identification of promising, high-reward subgoals (tokens, sections of text, etc.) within a response with (ii) aggregation of promising subgoals into subsequent prompts. The proposed method is loosely based on the standard model predictive control (MPC) formulation, with the MPC-based formulation of Section 4.1 serving as the point of departure; Principles 1 and 2 of Section 4.2 are the core components of the approach, however, and these are not clearly MPC-based. Experimental evaluation of TPMC against relevant baselines on machine translation, long-form response generation, and program synthesis problems is provided. The experimental evaluation indicates that TMPC outperforms baselines on two-thirds of the machine learning tasks, outperforms all baselines on the long-form response generation task, and achieves higher pass rates than all baselines on the program synthesis tasks.

**Strengths:**

The test-time alignment problem has seen immense interest from the community in recent years, so the topic of the paper is timely. The proposed TMPC approach is an intuitively appealing and natural approach to this problem, so the paper is likely of interest to the community. Drawing inspiration from MPC through the formulation given in Section 4.1 provides motivation and technical clarity for the (non-MPC) specifics of the proposed approach described in Section 4.2. The experiments consider three well-known benchmark problems and compare the proposed approach with a good variety of appropriately chosen, well-known baseline methods. The experimental results support the effectiveness of the proposed TMPC approach.

**Weaknesses:**

1. The connection to MPC feels somewhat overstated. Section 4.1 provides enough context to justify describing the approach as loosely MPC-inspired. However, since the main elements of the method are the non-MPC components outlined in Principles 1 and 2 (Section 4.2), framing as an MPC-based approach may be a bit strong.
2. The experimental results, though promising, contain some drawbacks that are not adequately discussed. First, though it is claimed on lines 401-402 that "TMPC consistently outperforms all test-time alignment baselines" on the machine translation tasks, Table 1 shows that TMPC is in fact outperformed by TPO on the Chinese-to-Russian translation task. In addition, as mentioned in footnote 3 on page 8, on the long-form response task TPO was stopped after two iterations instead of the four iterations used elsewhere due to implementation and/or hardware issues, not problems inherent to TPO. For this reason, the comparison results presented in Figure 3 are incomplete with respect to TPO; it is especially important to have an accurate picture of TPO's performance given that TPO was a top competitor on the machine translation tasks.

**Questions:**

1. In what way is the formulation of the test-time alignment problem as a sequential decision-making problem (lines 97-98) novel?
2. By what mechanism does the does the design of TMPC "achieve a better balance between accurate credit assignment and the size of the search space" (lines 240-241)?
3. Is it true that Principles 1 and 2 on page 5 are not clearly related to MPC?
4. Can you elaborate on the fact that GPT-4o outperforms or is highly competitive with TMPC in the experiments presented in Table 1?
5. Can you comment on the fact that on lines 401-402 it is stated that "TMPC consistently outperforms all test-time alignment baselines" on the machine translation tasks, yet Table 1 shows that TMPC is outperformed by TPO on one of the three tasks?
6. As mentioned in footnote 3 on page 8, on the long-form response task TPO was stopped after two iterations instead of four iterations; does this negatively impact the fairness of the comparison between TPO and TMPC in Figure 3?

---

> ### Author Response · Authors · 2025-11-20
> **Response to Reviewer UYVg (1/2)**
>
> We thank the reviewer for the thoughtful and constructive feedback. We address each concern point-by-point below and have revised the paper accordingly.
>
> ---
>
> ### [W1, Q3] Relationship between Principles 1 & 2 and MPC
>
> > *Since the main elements of the method are the non-MPC components outlined in Principles 1 and 2 (Section 4.2), framing as an MPC-based approach may be a bit strong.*
>
> Conceptually, TMPC is a **task-adapted instance of MPC** applied to LLM test-time alignment. In classical MPC, one repeatedly solves a **finite-horizon optimal control problem** over a planning horizon (H) and executes only the first portion of the optimized control sequence before re-planning. In Section 4.1 we follow exactly this structure: given a state (s_t), we solve a local trajectory optimization problem over horizon (H), then “execute” only part of that plan before moving the horizon forward.
>
> In TMPC, what is adapted to text generation is **not the MPC loop itself**, but what counts as an “action” and how we represent “executed” parts of the trajectory.
>
> Principles 1 and 2 are precisely what make this MPC loop usable for LLM preference alignment:
>
> * **Principle 1 (Hindsight Subgoal Identification)** says *which parts* of past rollouts should be treated as “executed” in the MPC sense: we score previously generated segments, pick the high-reward ones, and insert them into ($\mathcal{B}$).
> * **Principle 2 (Subgoal-Conditioned Re-Generation)** specifies *how* to re-plan from those executed parts: each new short-horizon trajectory is generated **conditioned on the buffer**, i.e., we repeatedly solve a local optimization problem over trajectories that extend or refine these committed subgoals.
>
> Thus, TMPC remains MPC in spirit: it **iteratively solves finite-horizon problems, executes only a subset of the plan, and then re-plans from an updated state**.
>
> Intrestingly, this perspective is also supported by cognitive science: humans naturally decompose long-horizon tasks into intermediate subgoals to reduce planning complexity rather than optimize every primitive action individually. For example, [1] shows that human planners choose subgoals that reduce cognitive load and narrow the future search space.
>
> We have also updated Section 4.2 (lines 305-311) to highlight the connection between TMPC and the classic MPC.
>
> [1] Binder, Felix Jedidja, et al. "Humans choose visual subgoals to reduce cognitive cost." Proceedings of the annual meeting of the cognitive science society. Vol. 45. No. 45. 2023.
>
> ---
>
> ### [W2, Q5] Misstatement: “TMPC consistently outperforms all test-time baselines”
>
> > *On lines 401–402 it is stated that “TMPC consistently outperforms all test-time alignment baselines” on the machine translation tasks, yet Table 1 shows that TMPC is outperformed by TPO on one of the three tasks.*
>
> We thank the reviewer for catching this. The sentence was indeed a misstatement and has been corrected.
>
> For zh→ru, TPO slightly outperforms TMPC on $SEGALE(_\text{comet})$, we have updated the text (Section 5.4, lines 413-414) to:
>
> > “TMPC achieves the best or second-best performance across all translation directions.”
>
> ---
>
> ### [W2, Q6] Fairness of TPO vs. TMPC
>
> > *As mentioned in footnote 3 on page 8, on the long-form response task TPO was stopped after two iterations instead of four iterations; does this negatively impact the fairness of the comparison between TPO and TMPC in Figure 3?*
>
> We appreciate this concern about fairness and have clarified both the **cost accounting** and **empirical behavior** of TPO in the revised version.
>
> 1. **Cost-matched comparison.**
>    In our implementation, each TPO iteration:
>
>    * samples 3 candidate responses, and
>    * requires one textual loss and one textual gradient computation per candidate.
>
>    Concretely, each iteration therefore requires (2 + 3 = 5) LLM calls. Two iterations correspond to **Best-of-10** in terms of total calls, which matches TMPC’s budget (initial response + 3 iterations × 3 rollouts = 10 responses). We also ran TPO with **4 iterations** (equal to Best-of-20 cost, i.e., 2×TMPC) with two A6000 GPUs.
>
> 2. **Observed performance vs. iterations.**
>    On HH-RLHF, the average reward (higher is better) was:
>
>    | Method / Iterations | 2 iters | 3 iters  | 4 iters |
>    | ---| ------- | -------- | ----- |
>    | TPO | 4.19 (10 LLM calls)    | – | 4.39 (20 LLM calls)  |
>    | TMPC| 4.53 (7 LLM calls)    | 4.60 (10 LLM calls)  | –  |
>
> We observe that TPO does improve with more iterations and can slightly exceed Best-of-20 at 4 iterations, but this requires roughly twice the number of LLM calls compared to TMPC. In contrast, TMPC reaches a higher reward with only 3 iterations (10 generations in total).
>
> TMPC is also more sample-efficient: with only 2 iterations, TMPC already surpasses the performance of TPO at 4 iterations. This further confirms TMPC’s advantage in efficiency. We have added TPO’s 4-iteration performance to Figure 3 of the revision.

---

> > ### Author Response · Authors · 2025-11-20
> > **Response to Reviewer UYVg (2/2)**
> >
> > ### [Q1] Novelty of formulating test-time alignment as sequential decision making
> >
> > > *In what way is the formulation of the test-time alignment problem as a sequential decision-making problem (lines 97–98) novel?*
> >
> > We thank the reviewer for this opportunity to further highlight our novelty and contributions. Our contribution is not just introduce the idea that language generation can be treated as a sequential decision process. Rather, our novelty lies in:
> >
> > 1. **Using the sequential decision-making view to unify and analyze existing test-time alignment methods.**
> >
> >    * guided decoding corresponds to **token-level actions**, suffering from a **curse of horizon**, and
> >    * naive iterative refinement corresponds to **response-level actions**, suffering from a **curse of dimensionality**.
> >
> > 2. **Recasting test-time alignment as a trajectory optimization problem solved with MPC-style planning.**
> >    We explicitly formulate test-time alignment as a **trajectory optimization problem**. Principles 1 and 2 operationalize this trajectory-optimization view via subgoals and subgoal-conditioned rollouts.
> >
> > To the best of our knowledge, no prior test-time alignment method both (a) frames token-level and response-level methods under a unified trajectory-optimization lens, and (b) introduces MPC as a planning mechanism in this setting, which, to our knowledge, has not been explored before.
> >
> > Our contributions in Section 1 (Lines 100-103) has been updated to reflect this clarification.
> >
> > ---
> >
> > ### [Q2] How TMPC balances credit assignment vs. search space size ?
> >
> > > *By what mechanism does the does the design of TMPC "achieve a better balance between accurate credit assignment and the size of the search space" (lines 240-241)?*
> >
> > We appreciate the opportunity to clarify this point. TMPC achieves this balance through **two  mechanisms**:
> >
> > 1. **Subgoal-level evaluation shortens the credit horizon.**
> >    Instead of assigning reward across hundreds of tokens (guided decoding) or treating the entire response as one action (response-level rewriting), TMPC evaluates **mid-sized segments** discovered via hindsight. Rewards therefore propagates over a much shorter trajectory, making credit assignment substantially more stable.
> >
> > 2. **The subgoal buffer ($\mathcal{B}$) restricts the search space to high-quality regions.**
> >    Rather than searching over all possible full responses, TMPC **anchors each iteration on previously validated high-reward segments** stored in ($\mathcal{B}$). This forces exploration to occur in neighborhoods of promising partial solutions, dramatically shrinking the effective action space.
> >
> > Together, these two mechanisms give TMPC a search space that is **much smaller than naive refinement** but with **much better credit accuracy than token-level guidance**, enabling more stable and efficient improvement.
> >
> > Section 4.1 (lines 259-261) has been updated to emphasize this operational mechanism directly.
> >
> > ---
> >
> > ### [Q4] Why is GPT-4o competitive or stronger than TMPC?
> >
> > > *Can you elaborate on the fact that GPT-4o outperforms or is highly competitive with TMPC in the experiments presented in Table 1?*
> >
> > We appreciate the opportunity to clarify the role of GPT-4o in our experiments.
> >
> > * GPT-4o is a much larger proprietary model, trained with substantial resources and likely with extensive preference alignment. In our setup it serves as a realistic upper bound for our 8B-parameter open-source backbone (LLaMA-3.1-8B).
> > * Our focus is not on “beating GPT-4o” across the board, but on showing that test-time alignment alone can substantially close the gap between a small open-source model and a state-of-the-art proprietary model.
> >
> > On zh→en, which is arguably the easiest direction for many models, TMPC-aligned LLaMA-3.1-8B-Instruct slightly exceeds GPT-4o on $SEGALE(_\text{comet})$ (94.62 vs. 94.58), illustrating that, for directions where the backbone is already strong, TMPC can **amplify existing capabilities** to the point of rivaling a much larger model.
> >
> > We have added a brief discussion in the Conclusion section (lines 535-537) emphasizing that:
> >
> > > TMPC is ultimately bounded by the capabilities of its backbone model; test-time alignment setting cannot create abilities the model lacks, but TMPC can reliably amplify existing strengths (e.g., Chinese–English translation), sometimes enabling smaller models to match or exceed much larger ones.

---

### Author Response · Authors · 2025-11-20
**General Response**

Dear reviewers,

Thank you for your thoughtful and constructive feedback during the review process. We appreciate the time and effort that went into the evaluations and the helpful suggestions for improving our submission.

Dear (new) AC,

We also appreciate the AC's effort in taking over the assessment of our submission. Below, we provide a brief summary of the improvements made during the rebuttal period, with all changes highlighted in blue in the revised manuscript.  We are happy to clarify any remaining questions in the final days of the rebuttal process.

---

### **(1) Additional empirical analyses**

Reviewer 1rTe suggested evaluating the sensitivity and robustness of TMPC. In response, we added:

* **Threshold sensitivity (Table 2(c); Section 5.5).**
We examine how the buffer threshold ( \alpha ) influences performance. Results show that TMPC remains stable across a wide range of values. We also explain the behavior when ( \alpha ) is too low (early admission of weak segments) or too high (reduced diversity approaching Best-of-(N)). Reviewers also asked how ( \mathcal{G} ) affects outcomes. We clarify its role and report the result.

* **Ablation of TMPC’s two principles (Table 2(d); Section 5.5).**
For TMPC's two principles contribution, we ablate (i) hindsight subgoal identification and (ii) subgoal-conditioned rewriting. Both contribute meaningfully, and the full TMPC design performs best.

* **Compute and latency (Appendix D.3).**
We added wall-clock latency and throughput comparisons across all guided decoding, TPO (2 iterations and 4 iterations) and TMPC.

### **(2) Fair comparison to TPO (Section 5.4; Figure 3)**

Reviewer UYVg raised the concern that stopping TPO after two iterations in HH-RLHF might affect fairness. In the revision, we now include TPO at four iterations (which requires two RTX A6000 GPUs to avoid out-of-memory issues) and report the complete compute cost for all methods.

The updated results show that TMPC still surpasses TPO even when TPO is given more compute. Section 5.4 has been updated to reflect these findings, and Figure 3 now includes the full comparison.

### **(3) Concrete subgoal examples and coherence analysis (Appendix G)**

Reviewer SBgL and Reviewer 1rTe suggested providing clearer examples of the subgoals that TMPC identifies, as well as illustrating how coherence is preserved when these subgoals originate from different rollouts

We now include real examples from MT, HH-RLHF, and program synthesis, showing buffer construction and subgoal retention.
To keep the main text concise, we illustrate the mechanism with compact, hand-selected responses in our responses, while appendix contains the full long-form outputs.

We also clarify that TMPC do not concatenate subgoals: TMPC generates a fresh full response conditioned on high-reward segments. Even when sampled subgoals conflict, outputs remain coherent.

### **(4) Paragraph-Level MT Heuristic Ablation (Appendix D.2)**

We added the heuristic method (fixed sentence boundaries) suggested by Reviewer SBgL, showing that even with natural boundaries, hindsight identification yields better translations.

### **(5) Expanded related work on subgoal planning (Section 2.3)**

Reviewer FkBi noted missing connections to prior subgoal-based methods. We now discuss some subgoal generation work for LLM/VLM-based tasks, and hindsight goal-generation in RL, and contrast them with TMPC's test-time, frozen-LLM setting and fixible subgoal formulation.

---

### Meta-Review · Area_Chair_987A · 2026-01-02

**Summary:**

The paper proposes Textual Model Predictive Control (TMPC), a novel framework for aligning large language models (LLMs) with human preferences at test time by casting the generation process as a sequential decision-making problem. The framework is evaluated on long-form machine translation, long-form response generation, and program synthesis, demonstrating improvements over several training-time and test-time baselines.

**Reviewer Concerns:**

1. The connection to MPC feels somewhat overstated
2. Missing comparisons to sequence-level rewriting baselines
3. Underspecified subgoal & buffer aggregation

**Reviewer Scores:**

The paper received all positive ratings. There are some minor concerns before the rebuttal. The authors have addressed all of them. AC made the acceptance decision.

---

### Decision · Program_Chairs · 2026-01-26

Accept (Poster)